
# Developing European operational oceanography for Blue Growth, climate change adaptation and mitigation and ecosystem-based management

J. She[1], I. Allen[2], E. Buch[3], A. Crise[4], J. A. Johannessen[5], P. Y. Le Traon[6], U. Lips[7], G. Nolan[4], N. Pinardi[8], J. H. Reißmann[9], J. Siddorn[10], E. Stanev[11], and H. Wehde[12]

[1]Department of Research, Danish Meteorological Institute, Copenhagen, Denmark
[2]Plymouth Marine Laboratory, Plymouth, UK
[3]EuroGOOS AISBL, Brussels, Belgium
[4]Istituto Nazionale di Oceanografia e di Geofisica Sperimentale, Trieste, Italy
[5]Nansen Environmental and Remote Sensing Center, Bergen, Norway
[6]Mercator Ocean and Ifremer, Ramonville St Agne, France
[7]Marine Systems Institute, Tallinn University of Technology, Tallinn, Estonia
[8]Department of Physics and Astronomy, Alma Mater Studiorum University of Bologna, Italy
[9]Bundesamt für Seeschifffahrt und Hydrographie, Hamburg, Germany
[10]Met Office, Exeter, UK

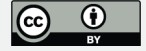

[11]Department of Data Analysis and Data Assimilation, Helmholtz-Zentrum Geesthacht, Hamburg, Germany

[12]Institute of Marine Research, Bergen, Norway

Received: 26 October 2015 – Accepted: 17 November 2015 – Published: 21 January 2015

Correspondence to: J. She (js@dmi.dk)

Published by Copernicus Publications on behalf of the European Geosciences Union.

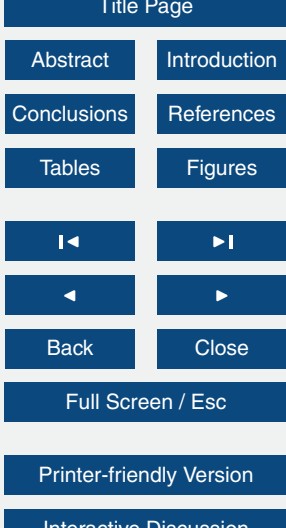

**OSD**

doi:10.5194/os-2015-103

**Developing European operational oceanography for Blue Growth**

J. She et al.

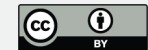

## Abstract

"Operational Approaches" have been more and more widely developed and used for providing marine data and information service for different socio-economic sectors of the Blue Growth and to advance knowledge about the marine environment. The objective of operational oceanographic research is to develop and improve the efficiency, timeliness, robustness and product quality of this approach. This white paper aims to address key scientific challenges and research priorities for the development of operational oceanography in Europe for the next 5–10 years. Knowledge gaps and deficiencies are identified in relation to common scientific challenges in four EuroGOOS knowledge areas: European Ocean Observations, Modelling and Forecasting Technology, Coastal Operational Oceanography and Operational Ecology. The areas "European Ocean Observations" and "Modelling and Forecasting Technology" focus on the further advancement of the basic instruments and capacities for European operational oceanography, while "Coastal Operational Oceanography" and "Operational Ecology" aim at developing new operational approaches for the corresponding knowledge areas.

## 1 Introduction

Operational oceanography, including ocean monitoring, analysis, reanalysis, forecasting and service provision is a branch of science that requires continuous implementation of the most advanced research findings to comply with ocean user needs. Inherent to operational oceanography is also the sustained production, timely delivery, automated qualification and free access to observations in near real time. Moreover, operational oceanography delivers products and information that are crucial for the research community to gain major understanding and advance knowledge and technology in the marine sector.

Discussion Paper | Discussion Paper | Discussion Paper | Discussion Paper |

**OSD**

doi:10.5194/os-2015-103

**Developing European operational oceanography for Blue Growth**

J. She et al.

Title Page

Abstract | Introduction

Conclusions | References

Tables | Figures

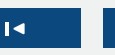 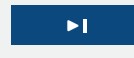

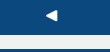 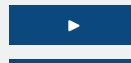

In the past decades, due to growing blue economy and challenges for adaption and mitigation to climate change as well as the improved capacities on operational marine service, "Operational Approaches" have been developed for a variety of socio-economic sectors, ranging from public service for coastal hazards in the beginning to emerging areas such as marine ecosystem and maritime services and integrated coastal zone management services. Such Operational Approaches share common features in their value-chain, i.e., user- and science-driven, knowledge- and technology-based, operation-practiced and service-oriented (She, 2015). The areas of interests for future research are determined by both user needs and current state of the scientific frontier. New knowledge and technologies, generated from the research, will then be incorporated into operational oceanography systems that provide the users with data and information products.

Operational oceanography in Europe was mainly initiated and sustained at national level before the 1990s. Aiming at integrating the operational oceanography development in regional and European scales, EuroGOOS from its very start established Regional Operational Oceanography Systems (ROOSes), such as for the Arctic Ocean, Baltic Sea, Northwest Shelf Sea, Ireland–Biscay–Iberia Seas and the Mediterranean Sea, EuroGOOS and its ROOSes have played an active role in data exchange, sharing the best practice and knowledge, harmonising monitoring networks and forecasting systems and stimulating joint research activities. Since Framework Program IV, the European Commission (EC) has continuously supported research on integration and development of European operational oceanography monitoring and forecasting systems, especially through Operational Forecasting Cluster projects (Cieslikiewicz et al., 2004), MERSEA (Marine Environment and Security for the European Area, Johannessen et al., 2006) and GMES (Global Monitoring for Environment and Security, currently referred to as Copernicus) Marine Service program (Bahurel et al., 2010). The development in the last 20 years has helped advance the existing national services and establishing new ones in many of the European countries. At the European level, an integrated capacity – the MyOcean operational monitoring and forecasting systems

Discussion Paper | Discussion Paper | Discussion Paper | Discussion Paper |

**OSD**

doi:10.5194/os-2015-103

**Developing European operational oceanography for Blue Growth**

J. She et al.

**OSD**

doi:10.5194/os-2015-103

**Developing European operational oceanography for Blue Growth**

J. She et al.



for global, Arctic and European regional seas has been established, which is now transformed into the Copernicus Marine Environmental Monitoring Service (CMEMS, http://marine.copernicus.eu/) program in the period 2015–2020.

Thanks to these national- and EU-funded programs we have seen major scientific achievements in the development of Earth Observation (EO) data management, short-term forecasting systems (including data assimilation) and reconstruction of long-term historical database through reanalysis and reprocessing. Long-term prediction, ecosystem prediction, coastal services and optimisation of European marine monitoring systems, have also been improved but with relatively lower levels of maturity and integration than the physical part of the CMEMS system.

In recent years, user requirements for operational marine data and information have largely increased due to the growing blue economy (e.g. marine energy, maritime transport, coastal and offshore engineering and marine bio-resources), implementation of European polices in marine-related Directives and regional marine environmental conventions (e.g. ecosystem-based management), adaptation to and mitigation of climate change as well as public services (e.g. disaster warning and protections). Although European operational oceanography has made significant advancements in the last two-decades, great challenges still exist in view to serve fast growing user needs. A large part of them can be summarised in four key knowledge areas: (i) European ocean observations, (ii) modelling and forecasting technology, (iii) operational oceanography in the coastal oceans and (iv) Operational Ecology (OE) (She, 2015).

This paper describes the objectives, challenges and research priorities in the above four areas, both in the short- to mid- term (1–5 years) and long-term (5–10 years and more). Among the four areas, (i) and (ii) focus on the further advancement and integration of existing operational oceanography areas. The two areas are closely integrated and provide a basis for building up European operational oceanography, which will be described in the Sects. 2 and 3. (iii) and (iv) are identified as two of the major emerging operational oceanography areas where the operational approaches based on the scientific state-of-the-art are still under development and which have to increase

Discussion Paper | Discussion Paper | Discussion Paper | Discussion Paper |

the significance in supporting sustained socio-economic development. Such an operational approach will provide a sustained development and service platform and significantly improve efficiency, quality and timeliness of the current services supporting Blue Growth, especially for the implementation of integrated coastal zone management and ecosystem-based management. The research in (iii) and (iv) can benefit from (i) and (ii), but also develop in their own directions as emerging research areas. Details can be found in Sects. 4 and 5. It is notified that the areas (iii) and (iv) are partly overlapping with (i) and (ii) but with different focuses and ambitions. A summary and discussion is given in Sect. 6, to provide a harmonised overview and address some missing issues of the paper.

## 2  European Ocean Observations

Since the establishment of EuroGOOS, it has been a central focal issue of EuroGOOS research to sustain, enhance and optimise the European ocean observing systems (Prandle et al., 2003; Nittis et al., 2014). With dual roles in ocean monitoring, i.e., both as observation providers and users, EuroGOOS members have different concerns. As a data provider, one needs to maximise the value of end-to-end data delivery and improve the cost-efficiency for making observations; as a user, one requires easy, fast and open access to a maximum of available qualified observations for operational oceanography applications.

Maximising the value delivery: as monitoring agencies, EuroGOOS members are responsible for delivering observations with maximised benefits to users for supporting European Blue Growth and public affairs:

– Values from data to product: improving observational data use for core marine products through (i) the timely delivery of available observations for operational use, (ii) the maximum use of observations in analysis, forecast, reanalysis and reprocessing, (iii) improved understanding of product skill through improved use of observations in validation and verification activities.

Discussion Paper | Discussion Paper | Discussion Paper | Discussion Paper |

**OSD**

doi:10.5194/os-2015-103

**Developing European operational oceanography for Blue Growth**

J. She et al.

**OSD**

doi:10.5194/os-2015-103

**Developing European operational oceanography for Blue Growth**

J. She et al.

- Values from data to knowledge: new knowledge generation by using observations together with models to understand physical and ecosystem processes and improve model parameterisations/forecasts.

- Values from data to socio-economic benefit: exploiting societal value of marine observations through innovative fit-for-purpose socio-economic applications in a variety of social benefit areas by using observations together with models and sectorial data.

Improving the cost-efficiency: EuroGOOS members need to undertake cost-efficient monitoring activities. This requires research and development on the assessment and design of cost-effective ocean observing networks through optimisation of sampling strategy, integration and coordination of observational infrastructure and efficient data management.

Data access and harvesting for operational oceanography applications: EuroGOOS needs to quantify the needs of ocean observations for operational oceanography applications, including parameters, data quality, sampling density and delivery time window. This analysis is instrumental to produce a coherent vision on future development of the observational component and its research and innovation priorities. In addition, timely access to the observations, both in online and offline modes, must be ensured. This requires EuroGOOS to work closely with other European ocean monitoring and data providers and management centres. Among the former are the environmental monitoring agencies coordinated under regional conventions (Helsinki Convention, Oslo and Paris Convention, and Barcelona Convention) and EEA, fishery monitoring community and research and commercial monitoring communities. Data management centres include ICES for handling marine and ecosystem data from the Baltic and North Sea, SeaDataNet for managing the offline physical and biogeochemical data, the CMEMS In-Situ Thematic Assembly Centre (TAC) for real time and delayed mode data required by the CMEMS and EMODnet for managing all types of marine data ranging from physical data to human activities, both online and offline. All these initiatives should

be further coordinated. EuroGOOS members are directly involved in EMODnet and the CMEMS in situ TAC and this ensures that these two major initiatives contribute to the overarching goal of facilitating the access to ocean data for operational oceanography. EuroGOOS also has a vision on observing systems for establishing a close dialogue with major users (e.g. COPERNICUS Marine Services) in order to align efforts to their requirements (and take advantages of feedbacks) and at same time to influence/harmonize the development of the national components.

Operational monitoring and data handling in emerging areas: our knowledge on marine ecosystems are evolving in the process of serving the growing blue economy and ecosystem-based management, and new challenges are also identified for data and information needs in emerging areas. Such emerging areas include, but not limited to, bottom sedimentation and resuspension, ocean acidification, marine pollution in related to noise and marine litter especially plastic/paraffin etc. These areas are normally beyond the existing scope of operational oceanography hence new monitoring and modelling technology should be developed. Furthermore, it becomes increasingly important to integrate "non-operational" observations, e.g., from tagged marine mammals, offshore commercial platforms and research observatories as well as sectorial information, e.g. ship data from Automatic Identification System, into an operational monitoring and data management framework.

Research on European Ocean Observations will aim at delivering the above objectives. The basic aspect of this research is to integrate existing observational infrastructure in operational oceanography. As emphasized in the EuroGOOS Strategy Plan (2014–2020), (Nittis et al., 2014), EuroGOOS will promote the need for the development of an integrated European Ocean Observing System (EOOS) during the coming years in partnership with the EuroGOOS ROOSes. The proposed system will be based to a large extent on past and planned investments: national systems, regional collaborative observing programs such as FerryBox and Voluntary Observing Ships, European programs and research infrastructures such as: Euro-Argo, JERICO-NEXT, FixO3, EGO, HF-Radars etc. However, following a system approach implies an addi-

tional level of operational networking and a governance scheme that will allow common programing and joint investments.

EuroGOOS is taking the initiative to lead and coordinate activities within the various observation platforms by enhancing the ROOS cooperation and establishing a number of Ocean Observing Task Teams such as HF-Radar, Glider, Ferrybox and Tide Gauges etc. and with strong link to Euro-Argo and its European legal entity (Euro-Argo ERIC). The purpose is to get these groups well organized creating synergy within the Task Teams themselves and across the Task Teams. This effort will be carried out in collaboration with the European Marine Board and other initiatives such as JPI-Oceans.

The development of satellite oceanography in the last two decades has also become a major component of operational oceanography as documented by Le Traon et al. (2015). Satellites provide real time and regular, global, high spatial and temporal resolution observation of key ocean variables that are essential to constrain ocean models through data assimilation and/or to serve downstream applications.

The future research on European ocean observations will evolve with advances in the observation capacities, such as the variety of Argo profiling floats (e.g. Bio-Argo, shallow water-Argo, abyssal-Argo, under-ice Argo), innovative in situ monitoring (e.g. ITP – Ice Tethered Profiler, Ice Mass Balance Buoys, ferrybox and gliders etc.), cabled observatories and ocean acoustics. Moreover, integration with satellite based observations, both polar-orbiting and geostationary satellites, are highly important. The outlook on future missions within the next decade is promising. The satellite constellation should be improved and new missions with a potentially large impact for operational oceanography (such as the Sentinel missions) should be demonstrated. International collaboration will be crucial to optimize and make best use of the satellite observations (e.g. sensor synergy, calibration, validation) from the growing number of space agencies. Moreover, more efforts will also be required to ensure homogenized and inter-calibrated data sets from multiple missions for all essential ocean variables.

The on-going and forthcoming EC Horizon 2020 supported projects such as AtlantOS for the Atlantic Ocean, JERICO-NEXT for coastal observatories, and the calls

**OSD**

doi:10.5194/os-2015-103

**Developing European operational oceanography for Blue Growth**

J. She et al.

Title Page

Abstract | Introduction

Conclusions | References

Tables | Figures

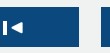 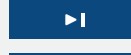

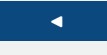 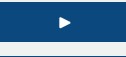



Discussion Paper | Discussion Paper | Discussion Paper | Discussion Paper |

on the Integrated Arctic Observing System and the Mediterranean Observing System with submission in February 2016 will strengthen the integration of European ocean observing systems.

In the long-run, it is foreseen that European Ocean Observations will become more integrated, coordinated and efficient. The related activities will be described below in two categories: development and integration of ocean observing systems and assessment and optimisation of observational networks. The former is dedicated to maximum value delivery of observations, ad hoc optimisation of monitoring networks and data harvesting for operational oceanography and the latter to improve the cost-effectiveness of the EOOS through quantitative impact and design studies.

## 2.1 Development and integration of ocean observing systems

The goals of the integration of the ocean observing systems are: (i) maximising the amount of timely and quality assured observations for operational oceanography, (ii) improving the cost-effectiveness of current monitoring components, (iii) improving the sustainability, (iv) delivery of new observations for operational oceanography and (v) improving the efficiency of managing and using big data. To reach these goals, the following challenges have been identified.

### 2.1.1 Short- to mid-term objectives

– Reducing the observation gaps: integrating existing non-operational, multi-source observations at regional level to ensure more timely access, delivery, and usage of observations for analysis/forecasting and regular ocean state estimation, identify critical "data delivery time windows" for operational forecasting and harmonise the data format, metadata and quality standard, integrating new observations into the existing operational data flow, promoting the historical data gathering in coordination with EMODnet (in particular for biogeochemical variables), widening the

**OSD**

doi:10.5194/os-2015-103

**Developing European operational oceanography for Blue Growth**

J. She et al.

Discussion Paper | Discussion Paper | Discussion Paper | Discussion Paper

usage of innovative cost-effective monitoring technology e.g. ferrybox, HF radar and Bio-Argo etc. in operational monitoring.

– Ensuring open availability of innovative multi-sensor satellite observation retrieval algorithms for essential ocean and ice variables with higher quality: using in situ measurements and multi-variate met-ocean data to calibrate, validate and improve the relevant remote sensing data and products, including possible new products derived from space infrastructures both in Europe and other countries such as USA, China, Japan and India etc.

– Coordinated use of marine infrastructures at regional level: for instance in multilateral coordination of research vessel based monitoring, mobilisation of additional relocatable observational infrastructure (e.g. AUVs, gliders and drifters) with coordinated sampling schemes etc. Although difficult, coordinated monitoring planning such as on ship time, sampling locations and mobilisation of the observational infrastructure can make significant improvements in terms of the cost and benefit.

– Testing the effectiveness of existing (semi)automated sensors for chemical and biological observations.

– Data processing: further development of real-time quality control protocols, development of advanced data products (value-added) merging different type of observations, especially those including new satellite and in situ observations, establishing systematic and consistent observation-based analyses framework as suggested by Chapron et al. (2010).

### 2.1.2 Long-term objectives

– New observations: filling the monitoring gaps in key locations by deploying innovative multi-platform sensors, promote the development of a deep-sea network of

Discussion Paper | Discussion Paper | Discussion Paper | Discussion Paper | Discussion Paper |

**OSD**

doi:10.5194/os-2015-103

**Developing European operational oceanography for Blue Growth**

J. She et al.

pressure gauge (needed also for calibration of satellite sea level products), developing limited number of supersites located in critical areas (in particular in open sea) with a multi-platform approach, developing marine mammal tagged observations, developing operational monitoring instruments and data handling tools for underwater noise and marine litter.

– Integration of observations from the research community and private sectors: with the progress of engaging research community (e.g. promoting the use of data doi) and private sectors in operational oceanography, the observations made by them should be collected and shared for operational oceanography research and other secondary uses.

– Coordinated and cost-effective deployment of multi-platform infrastructure at regional level, e.g. high quality ship-board and bottom-mounted ADCP monitoring, ferrybox, HF radar, moorings, cabled stations, innovative use of light houses and other offshore platforms etc.

– Transferring, expanding and integrating mature, cost-effective monitoring technology e.g. HF radar for general operational use.

– New technology for operational monitoring: developing cost-effective multi-sensors and robust calibration protocol especially for biogeochemical measurements, sediment, underwater noises and marine pollutants.

– Exploring the operational potential of present and innovative initiatives in the field of citizen science (sea state observation, marine litter, ocean colour, jellyfish, etc.).

– Efficient big data management: it has been a challenge to quickly access and extract increasing amounts of Earth Observation (EO) data which can be of order of Peta- to Exabyte scale. The Earth System Grid Framework has been developed to facilitate data extraction from multiple data centres. However bottlenecks exist inside each data centre for online access to medium amounts of data ($10^2$–$10^3$ Tb).

Discussion Paper | Discussion Paper | Discussion Paper | Discussion Paper | Discussion Paper |

**OSD**

doi:10.5194/os-2015-103

**Developing European operational oceanography for Blue Growth**

J. She et al.

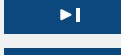

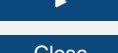

An efficient data management framework should be developed for online access, download, view and analysis to data from a distributed multi-server local network. Novel technologies will be foreseen to move toward an open source array-oriented database management system. Further development of data mining and image processing techniques is needed to facilitate the automatic extraction and analysis of patterns from big data sets.

– Interoperability: identifying a strategy to move from the NetCDF, file transfer based, data exchange technology to the GEOSS philosophy (compliant where necessary to ISO standards) based on interoperable web services.

## 2.2   Assessment and optimal design of ocean observing networks

The goal of the marine monitoring network assessment and optimal design research is to identify the gaps in existing observing systems and to optimise their cost-effectiveness. The EC has continuously supported this research area since early 2000. The assessment and design studies can be divided into ad-hoc studies and quantitative studies. The ad-hoc studies have been carried out in many EC funded observing system projects such as EDIOS, SeaDataNet and recent fit-for-purpose assessment by DG-MARE (Sea Basin Checkpoint projects for European Seas). The ad hoc assessment work has led to the establishment of the meta database and identification of data availability and accessibility etc. On the other hand, a variety of quantitative assessment and optimal design research have also been carried out in EC projects ODON, ECOOP, JERICO and OPEC, and are now continuing in JERICO-NEXT and AtlantOS. Both statistical assessment and optimal design methods as well as assimilative model-based method – OSE (Observing System Experiment) and OSSE (Observing System Simulation Experiment), have been developed and applied in these projects. Large parts of the physical and biological operational monitoring network (SST, $T/S$, nutrients, oxygen and chl $a$) in European Seas have been assessed in terms of effective coverages and explained variance (She et al., 2007; Fu et al., 2011). The OSEs and

**OSD**

doi:10.5194/os-2015-103

**Developing European operational oceanography for Blue Growth**

J. She et al.

OSSEs have also been applied in assessing and optimising physical monitoring networks, e.g., in FP5 project ODON and FP7 project JERICO. The strengths of OSEs and OSSEs are that impacts of a given sampling scheme can be quantitatively assessed in terms of improvements of forecasts (Oke and Sakov, 2012; Turpin et al., 2015). The weakness is that the results are model dependent and it can only address one sampling scheme per simulation. The statistical method has the strength of being a quick assessment and can be easily applied to find one optimal sampling scheme among many given candidates. A potential integration of the two approaches is expected to combine the relative strong points.

### 2.2.1 Short- to mid-term objectives

– Quantitative assessment of gaps and redundancy for operational forecasting: assessing representativeness, sampling error and impacts of European marine monitoring in situ components (incl. non-operational components) on operational analysis and forecasting to identify critical gaps and redundancy areas, with including existing satellite data, modelling and assimilation techniques.

– Development of automatic observation network evaluation tools which can provides estimates of quality parameters of the network, such as effective coverage, sampling error, explained variance, reconstruction error and forecasting error, for sampling schemes defined by the users.

– Impact study method development: development of more robust methodologies (i.e. to ensure results as independent as possible from model and error assumptions) to conduct impact studies.

### 2.2.2 Longer-term objectives

– Optimal design: identification of critical observation gaps and redundancy in parameters, space and time, providing quantitative optimal designs of new cost-

**OSD**

doi:10.5194/os-2015-103

**Developing European operational oceanography for Blue Growth**

J. She et al.

effective components of EOOS as well as guidance to the in situ observing communities on how to optimise observing strategies (e.g., sampling scheme, technology etc.) and the complementarity with Sentinel missions, adopting an integrated, user-driven and science- and technology-based design approach by combining the relevant scientific, technological and management resources.

– Improvement of monitoring schemes at regional level: based on impact and/or design study, identifying monitoring cases with significant cost-effectiveness improvement in the integration of existing systems, ship time planning, integrated and/or mobilised use of observational infrastructures etc., implementing the cases by integrating monitoring technology (in situ and remote sensing), sampling schemes, monitoring objectives, modelling capacity, user needs and investment as a whole. Detailed knowledge should be developed on how different monitoring platforms, assimilation and understanding of dynamic processes can benefit each other to reach a cost-effective design of the system. Delivery time vs. user needs should also be mapped and evaluated for both physical and biogeochemical variables.

– Promote, design and carry out large scale, integrated field experiments: in order to make breakthrough in new areas of operational oceanography, such as for coastal shallow waters and operational ecology, dedicated large scale field experiments are needed with an integrated monitoring-modelling approach. The knowledge and technological gaps should be identified, filled and transformed into the corresponding monitoring and forecasting systems. Examples with more details can be found in Sects. 4 and 5 – Coastal Operational Oceanography Experiment and Operational Ecology European Experiment.

**OSD**

doi:10.5194/os-2015-103

**Developing European operational oceanography for Blue Growth**

J. She et al.

**OSD**

doi:10.5194/os-2015-103

**Developing European operational oceanography for Blue Growth**

J. She et al.

# 3 Operational modelling and forecasting technology

Modern ocean and ecosystem prediction and state estimation is built upon a combination of ocean models and observations. The advanced science and technology in forecasts is at the centre of earth system science challenges, as shown in Fig. 1, together with innovation, observing, responding and confining the impacts (ICSU, 2010). The accuracy of the ocean prediction relies on the model quality both on dynamics and numerical solver, model setup, quality and amount of forcing data and observation data and the quality of pre-processing, assimilation and post-processing technology. In this section we divided the modelling related research areas into model development and forecasting technology, e.g., data assimilation, nowcasting and probabilistic forecast etc.

## 3.1 Model development

In recent years seamless modelling and forecasting system development has become a major focus to develop a unified framework for modelling and forecasting on both weather and climate scales (Shukla, 2009). Recently the WMO published the scientific report "Seamless prediction of the earth system: from minutes to months" which announces a new era of development of our forecasting capacity into "Unified Earth System Models – UEM" (WMO, 2015). Some countries, such as the UK and USA, have worked on a seamless approach to weather and climate prediction by developing common modelling tools for weather and climate for years. For the ocean–sea ice-wave-ecosystem prediction, existing boundaries of prediction between different time scales were mainly delimited due to computational and model complexity considerations. Current CMEMS operational models such NEMO, HYCOM and HBM etc. have also been used in the long-term simulations such as hindcast, reanalysis and climate projections. It is timely to build the next generation European operational ocean–sea ice-wave-ecosystem models in the framework of the "Unified Ocean system Model (UOM)".

The UOM means that the ocean subsystem models (i.e., ocean, sea ice, wave, sediment transport, marine ecosystem etc.) are able to serve the purpose of applications on all time scales, ranging from nowcasting to climate projections. This requires that the model (i) has a high coding standard, flexible grid and efficient numerical schemes
to meet computational needs for both operational forecast and climate modelling, (ii) is able to properly resolve small scale features and extreme events as well as other features needed for operational services, (iii) meets the energy and mass conservation requirements for long-term simulations. The UOM should also be fully coupled between the subsystem models and the Unified Atmospheric Model (UAM).

Operational ocean modelling has been significantly advanced in the last 20 years in Europe. A great number of physical ocean-ice models have been developed and used in operational forecasting such as NEMO, HBM, HYCOM, ROMS, MITGCM etc. In recent years a very strong movement in the physical ocean modelling community is the NEMO model development, with supports from both the national and European level. More and more countries start to use NEMO as their operational model. On the other hand, using different models in Europe for operational forecasting are also necessary as no single model can solve all problems. Quite a few ecological models have also been developed for operational forecasting such as ERSEM, ERGOM, BFM, ECO3M, BIMS_ECO, NORWECOM, ECOSMO etc. High trophic models have also been developed for the forecasting purpose, e.g. in OPEC project. The state-of-the-art European wave models and ocean-wave coupling have been further advanced for operational forecasting in MyWave project, which is an important step towards Copernicus wave service.

There will probably be in the future several prototype European UOMs, depending on further development of the existing state-of-the-art and available resources (both funding and modelling expertise) in Europe. Some UOMs may have a capacity to cover a wide range of spatial scales ranging from coastal to global ocean. Others may only cover multi-basin, basin and coastal oceans.

**OSD**

doi:10.5194/os-2015-103

**Developing European operational oceanography for Blue Growth**

J. She et al.

The operational ocean models for the European Seas provide nowcasting and fore-casting ranging from hours to days, which have to resolve mesoscale and smaller scales, high frequency phenomena and extreme events. The models have to be cali-brated to reach certain quality standards to meet the user needs, and regularly verified against observations. These models have also been used for generating hindcast, re-analysis and climate projections. However, in order to use the existing operational mod-els for climate scale applications, there still exist significant challenges in improving the computing efficiency and energy and mass conservation features of the operational models. The benchmark test of the climate UOM should be made for above two is-sues.

The computational aspect of the UOM concerns both computation speed and to-tal consumption of electricity. Computational efficiency is the key both to enhance the speed and reduce the total energy consumption. Forecasting and climate modelling for the entire coupled ocean system in a probabilistic framework are extremely computa-tional demanding. For future seamless modelling, the minimum requirement is that the UOM should fulfil computational limits for both operational and climate modelling, e.g. delivering a 5–10 day forecast daily within 2–4 h and a hundred year run within a few months. In addition, the model code should be optimised in order to minimise the to-tal electricity consumption which needs close cooperation between model developers, HPC experts and hardware producers.

In order to use the operational UOM for climate applications, the model should be able to generate a stable solution (with no significant trend) by running for several hundreds of years without including anthropogenic effects. This serves as a basic re-quirement (of energy and mass conservation) for climate modelling. The development of UOM is a long-term goal which may be reached in 10 years or even longer, while the short- to mid-term model development will be mainly driven by large scale operational oceanography projects such as CMEMS and those in Horizon 2020 Calls which mainly focus on developing the existing modelling framework at basin and global scales. The

Discussion Paper | Discussion Paper | Discussion Paper | Discussion Paper |

**OSD**

doi:10.5194/os-2015-103

**Developing European operational oceanography for Blue Growth**

J. She et al.

ideal situation is that the short- to mid-term European ocean model development can be effectively integrated into the UOM framework.

In the short- to mid-term, the objective of the model development work is to develop a European UOM framework and continuously improve the deterministic prediction models with forecast range of 10 days or longer. The research should focus on (i) designing the UOM concept and framework and develop a roadmap towards the UOM, (ii) improving description of model processes so that each UOM sub-model can effectively model major features in the subsystem, (iii) improving the code quality and high performance computing, (iv) improving the UOM subsystem coupling and UOM-UAM coupling and (v) developing high resolution models with flexible grids and interfaces with basin and global scale models, and resolving coastal processes for downstream applications. Some of the above research topics, such as increased resolution, improved parameterisations and atmosphere–ocean–sea ice-wave coupling etc., have been addressed in the research priorities of CMEMS Service Evolution strategy (CMEMS STAC (Scientific and Technical Advisory Committee), 2015).

Modelling framework development: in the European ocean modelling community, a roadmap towards the UOM is needed, which shall cover but not be limited to, coding standards, code adaptation to many-core computer architectures, coupling framework, new model component e.g., sediment transport and high trophic level models and sharing best practices of the model development. Detailed analysis of user and computational needs on the future UOMs should be made. The best practices from both ocean and atmospheric model development should be used to develop such a roadmap.

Integration of best practice into the UOM framework: due to the lack of resources at national level for ocean model development it is very important to share best practice in operational modelling. One way for sharing best practice is through Community model development such as NEMO. There are also a few initiatives started recently to develop a Research to Operations (R2O) strategy meeting the needs for modernization of numerical models to support the forecasting process. One of such interesting platforms is the hurricane R2O developmental testbed (Bernardet et al., 2015), an initiative hing-

**OSD**

doi:10.5194/os-2015-103

**Developing European operational oceanography for Blue Growth**

J. She et al.

ing on three activities: establishing a solid code management practice, supporting the research community in using the operational model and inserting innovations and conducting model testing and evaluation in a well-established and harmonized framework. Such ideas, though applied in meteorology, can also be useful in the establishment of the UOM framework. Concerted action among the European modelling groups is also important for integrating the progress in the different modelling groups into the future UOM framework. EuroGOOS has initiated a Coastal and Shelf model Working Group (COSMO) to promote the model knowledge exchange and best-practice sharing.

Improving deterministic models: although operational physical ocean models are much more mature than the ecological models, there still exists well-known challenges such as unrealistic diapycnal mixing, resolving bottom layers and sharp pycnoclines, flow over steep topography, water exchange through narrow straits, configuration of surface fluxes in a coupled framework, vertical transport of substances, sub-grid parameterisation, binary identical code and capacity for using new high performance computing architectures. Progress in the above areas will directly improve the model quality.

Development of coupled systems: research in the development of the coupled system and predictability study will evolve in Horizon 2020 program and the Copernicus Service especially CMEMS systems. While coupled atmosphere–ocean-ice-wave models have been developed in global level for climate research and seasonal forecasting, regional coupled systems for synoptic scale prediction remain to be developed. Proper implementation of the air–sea–ice interaction and data assimilation for the coupled system are essential for correctly resolving corresponding diurnal variability. Predictability is expected to be prolonged in a coupled forecasting system, which should be explored. The future development will also contribute and draw momentum from on-going GODAE-OceanView (Brassington et al., 2015, in prep., https://www.godae-oceanview.org/publications/special-issues/).

Emerging modelling areas: It is worthwhile to mention that the model development in the CMEMS strategy mainly focuses on the evolution of the existing global and basin scale operational models (ocean–sea ice-wave-biogeochemistry), new emerging mod-

**OSD**

doi:10.5194/os-2015-103

**Developing European operational oceanography for Blue Growth**

J. She et al.

Discussion Paper | Discussion Paper | Discussion Paper | Discussion Paper |

els such as sediment transport and high trophic level models, and models for downstream services such as coastal inundation model, unstructured grid models have not been sufficiently addressed in the strategy. In addition to the model development, comprehensive verification studies should be made especially for the ecological models and Arctic models in order to understand the drawbacks of the models. For the ice model, mesoscale sea ice rheology will be needed to describe lead dynamics of the ice. More discussions on the development of marine ecosystem models can be found in Sect. 5 – Operational Ecology.

The above short- and mid-term research will significantly improve the efficiency and accuracy of the model performance at synoptic scales, which will provide a basis for building up European UOMs. In the long-term, it is important to reach breakthroughs in seasonal forecasting for the European earth system and to improve the quality and efficiency of the UOMs in generating climate simulations. The research here focuses on probabilistic forecast, coupled UAM-UOM models with multi-grids and medium-high resolution, efficient high performance computing for global, multi-basin and coastal scales. The research is a further extension and integration of the existing deterministic UAM-UOM modelling framework which has been developed in the short- and mid-term research.

In the long-term, UOMs for solving problems at pan-European Seas and Arctic-North Atlantic scale should be developed. Since European regional seas are connected through straits (some with widths of a few hundred meters to kilometres), the UOM for climate scale applications have to resolve such scales in order to model correctly the inter-basin transport. Besides, implementation of European policies, such as the Climate Directive, Common Fishery Policy and Marine Strategy Framework Directive etc., needs a harmonised European Sea database to support the decision-making. An UOM at pan-European scale will fit this purpose. The model system should be able to resolve and/or permit mesoscale eddies and resolve narrow straits. The current operational models, such as the UOM developed for deterministic prediction, can be further

**OSD**

doi:10.5194/os-2015-103

**Developing European operational oceanography for Blue Growth**

J. She et al.

Title Page

Abstract    Introduction

Conclusions    References

Tables    Figures

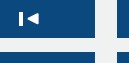    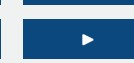

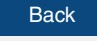    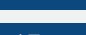

developed for this purpose with two-way nesting. Other alternatives include unstructured grid models.

For the seasonal and longer scales, it has been found that the Arctic condition has great impacts on the European weather and climate. An Arctic-North Atlantic coupled atmosphere–ocean-ice-wave component should be developed as a key part of the future European Earth System Model. The advantage of the regional coupled Arctic system is that high resolution can be used and research efforts can be focused on the Arctic related processes such as atmosphere–ocean-ice coupling and sea ice dynamics etc. A few regional coupled atmosphere–ocean-ice systems, e.g. RASM (Maslowski et al., 2012) and national systems in Sweden, Norway and Denmark, have already been tested for Arctic climate research. The development of the future Arctic-North Atlantic coupled model should also take the advantage of the Horizon 2020 Blue Growth Calls on Arctic: BG9 – Integrated Arctic Observing System and BG10 – Impact of Arctic on weather and climate in Northern Hemisphere and Europe, as well as the Year of Polar Prediction (YOPP).

As mentioned before, it is essential that the Climate UOM should be adapted to the multi-core and many-core supercomputing processors with efficient and balanced hybrid parallel computing. The current model code may have to be rewritten and restructured, as reported recently in the High Performance Computing workshops organised by ECMWF and NCAR. For example, stricter coding standard should be applied to ensure the run-to-run reproducibility. More efficient coding principle such as PSyKAl (Parallel System, Kernel and Algorithm), taken in the GungHo Project which is developing a new Dynamical Core suitable for the weather and climate simulations, may benefit the UOM development; upgrading the code with SIMD (Single Instruction Multiple Data, Poulsen et al., 2014) feature has proven the benefit for the model by using new vectorisation, efficient hybrid threading for multi-core and many-core architectures.

**OSD**

doi:10.5194/os-2015-103

**Developing European operational oceanography for Blue Growth**

J. She et al.

Discussion Paper | Discussion Paper | Discussion Paper | Discussion Paper |

OSD

doi:10.5194/os-2015-103

Developing European
operational
oceanography for
Blue Growth

J. She et al.

## 3.2 Forecasting technology

Advanced model code does not necessarily mean a good forecast. Initial and forcing errors are the two major sources of the forecasting error. There are normally two ways to deal with the initial error: one is assimilating observations to obtain a more realis-
5 tic initial field, the other is to perturb the initial field to generate ensembles which will be used to make a probabilistic forecast. The benefit of the ensemble forecast is that (at least) the white noise of the forecast can be largely removed by using ensemble mean, and the probabilistic forecast gives a valuable estimation of forecast uncertainties, furthermore the method enables possibilities for risk management. In this section
we focus on the future research on ocean data assimilation and ensemble forecasting technology.

### 3.2.1 Data assimilation

The reduction of the product uncertainties is a central challenge for operational modelling and services, which requires continuous innovations in data assimilation. Present
15 day assimilation approaches encompass a hierarchy of methods of increasing complexity, ranging from optimal interpolation to non-linear stochastic methods (CMEMS STAC, 2015). For open oceans, satellite measurements such as sea surface temperature, sea ice concentration and sea surface height and in situ observations of SST and $T/S$ profiles have been assimilated in global and regional forecasting systems for
the North Atlantic, Arctic and Mediterranean Sea, such as in CMEMS Marine Forecasting Centres. For coastal and shelf sea assimilation, there have been a number of successful stories, e.g. sea level assimilation in North Sea storm surge forecast (Zijl et al., 2013), SST assimilation in CMEMS NW shelf MFC and assimilation of SST, sea ice concentration and $T/S$ profiles in the Baltic Sea.
Major challenges in operational assimilation remain in the coastal and shelf waters for assimilating sea level both from satellite and in situ tidal gauges, surface currents from HF radar, ice thickness and ice drift as well as for assimilating biogeochemical

parameters. In this area, traditional Gaussian-distribution based assimilation methods such as 3-DVAR or Kalman Filter-based methods have shown improvements and potential for operational applications, such as in assimilating blended satellite-in situ sea level data in Baltic-North Sea in eSurge project, satellite chl *a* assimilation in OPEC project (http://www.marine-opec.eu/documents/deliverables/D2.6.pdf) and ferrybox SST/SSS/HF radar surface currents assimilation in the German COSYNA project (Stanev et al., 2013, 2015). However, technical difficulties remain, especially in cases with large spatial and temporal variations and high non-linearity, relatively large model uncertainties and insufficient real-time observations. All these factors, especially when added together, may often lead to non-Gaussian model error statistics which cannot be solved properly by traditional assimilation methods based on non-biased Gaussian distribution of error statistics. Severe model instability or unrealistic correction of the model initial fields may be generated.

New, innovative assimilation methods such as stochastic assimilation motheods and common data assimilation frameworks such as PDAF (Parallel Data Assimilation Framework) have been developed in the FP7 SANGOMA project. Independently of SANGOMA, other efforts on modular software development have also been initiated at other European institutions, such as the OOPS project at ECMWF. The following research and development activities on data assimilation are required:

**In the short- to mid-term**

– Common assimilation framework developments: development of community tools and diagnostics in observation space, sharing of assimilation tools with the ocean modelling community and observational experts, verification methods and inter-comparison protocols suitable to probabilistic assimilation systems.

– Transferring existing best practices into operational systems: calibrating and operationalising mature assimilation schemes for observations from research vessel, buoys, ferrybox, HF radar, altimetry and tidal gauges for coastal and shelf seas.

**OSD**

doi:10.5194/os-2015-103

**Developing European operational oceanography for Blue Growth**

J. She et al.



**OSD**

doi:10.5194/os-2015-103

**Developing European operational oceanography for Blue Growth**

J. She et al.

- – Development of new assimilation methods: stochastic assimilation methods, hybrid assimilation methods and assimilation methods addressing non-Gaussian error statistics.

- – Development of assimilation of new and novel observations: ice thickness, currents, nutrient profiles and plankton, new data assimilation methods designed to handle strongly nonlinear dynamics and semi-qualitative information from satellites.

**In the longer-term**

- – Further development of innovative assimilation methods: improving atmospheric forcing using available observations via the ensemble Kalman filter and smoother, non-Gaussian extensions for non-linear transformations of probability distributions to reduce data assimilation biases by more realistic stochastic models, development of hybrid data assimilation method, developing and implementing advanced techniques to assimilate data into coupled ocean-ice-wave-atmosphere model systems.

More details the above research priorities can be found in the CMEMS Scientific Strategy (CMEMS STAC, 2015).

### 3.2.2 Probabilistic forecasts and forecast uncertainty quantification

Risk assessment and management has been set as a standard requirement for many sea-going operations and policy making, which raises needs for probabilistic forecasts and estimation of the forecast uncertainties. Due to the lack of ocean observations, it is not easy to quantify the forecast uncertainties by comparing the model data with observations. One way to estimate the model product uncertainties is to use single model ensembles or multi-model (super-ensemble) forecasts. Through perturbing the initial state, the lateral and vertical boundary condition errors and/or the model shortfalls in

Discussion Paper | Discussion Paper | Discussion Paper | Discussion Paper |

a sufficiently large range, it is expected that an ergodic set of the forecast ensembles can be generated which contains the true solution (the truth) as a subset. In this case, a probabilistic forecast can be estimated from the ensemble and/or super-ensemble products according to different user requirements, e.g., probability of the significant wave height higher than 5 m within the next 24 h. The best estimate of the forecast and its spread can also be derived. With a Gaussian-distribution assumption, the spread can be used as an estimation of the forecast uncertainty. A framework of probabilistic forecast production, validation and application has been well established in meteorology but much less in oceanography. Operational oceanography is presently developing these methods for marine short term forecasting (Counillon and Bertino, 2009).

Probabilistic forecast for waves and physical ocean conditions has been developed and used in European operational oceanography in the last decade, both with ensemble and (multi-model) super-ensemble forecast. ECMWF has operated global ocean wave ensemble forecasting for some years. A regional Baltic-North Sea wave ensemble forecast has been put in operation in 2014 in the MONALISA2 project. Increasing use of ensemble data assimilation method also provides a natural platform for making 3-D ocean ensemble forecast. For the European Seas, multi-model water level prediction has been developed for European Seas in ROOSes and in the ECOOP project, and used for national storm surge forecasts since early 2000s (Perez et al., 2012). Further development of multi-model ocean forecasting system has been an active part of MyOcean and CMEMS (Golbeck et al., 2015).

However, essential challenges in the ocean ensemble/super-ensemble forecast remain: due to the insufficient coverage of all kinds of uncertainties when generating the forecast ensembles, that the ensembles often partly contain the truth and cannot form an ergodic set, inefficient generation of the ensembles often leads to convergence of the ensembles which makes this issue worse. Multi-model ensemble in a certain sense effectively increases the number of independent ensembles and has shown very good results in ensemble forecasts. Furthermore the ensembles may not be Gaussian distributed and non-biased. In order to get a proper estimation of forecast uncertainty,

**OSD**

doi:10.5194/os-2015-103

**Developing European operational oceanography for Blue Growth**

J. She et al.

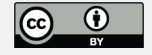

Discussion Paper | Discussion Paper | Discussion Paper | Discussion Paper

probability distribution function (pdf) based and bias-corrected uncertainty estimation should be developed and applied.

In the short- to mid-term, research is needed for establishing a framework for ocean model probabilistic forecast validation, building up probabilistic forecasts through advancing ensemble-based assimilation, improvement of ocean model ensemble generation with more effective perturbation of initial states, forcing, lateral boundary conditions and model shortfalls to get close to an ergodic set of ensembles, further development of multi-model ensemble forecasting and transferring to operations and advancing the ensemble/super-ensemble forecast by including real-time observations and Model Output Statistics (MOS) for forecast corrections.

It is obvious that seamless forecasting has to be treated in a probabilistic way for a fully coupled system. In the long-term, efficient methods should be developed for estimating the forecast uncertainty including bias correction and non-Gaussian distribution of the ensembles. With the Unified Earth System Models developed for the pan-European Sea and Arctic-North Atlantic scale, a probabilistic framework should be developed for seasonal forecasting and climate projections. The predictability study is needed to understand and assess the predictability of the ocean circulation, biogeochemistry and marine ecosystems at global, basin scale or regional scale, and to identify spatial and temporal scales with the strongest predictable signals in model system dynamic processes, initial states and forcing. For the historical data, the probabilistic framework and metrics are needed for the ocean reanalysis using ensemble techniques. Methods should be developed to ensure quality, homogeneity and robust uncertainty measures in the long-term time-series reconstructed from data or model reanalyses.

## 4 Operational oceanography in the coastal ocean

The coastal oceans, including coastal zones, offshore and open coastal waters, are important economic zones and key areas of European Blue Growth. One third of the

**OSD**

doi:10.5194/os-2015-103

**Developing European operational oceanography for Blue Growth**

J. She et al.

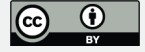

EU population lives within 50 km of the coast. The GDP generated by this population amounts to more than 30 % of the total EU GDP. The economic value of coastal areas within 500 m of the European shores has a total between EUR 0.5–1 trillion per annum (European Commission, http://ec.europa.eu/environment/iczm/state_coast.htm).

The coastal environment is experiencing its fastest changes ever recorded by instrumentally-sea level rise, coastal erosion, increasing water temperature and changing riverine inputs, water mass properties and mixing feature. The most vulnerable part of the coastal ocean is the Costal Shallow Waters (CSW) with a depth of a few tens of meters. This zone is subject to most dynamic changes made by winds, waves, tides, sediment transport, riverine inputs and human activities. They are also the hottest spots in marine spatial planning, maritime safety, marine pollution protection, disaster prevention, offshore wind energy, climate change adaptation and mitigation, ICZM (Integrated Coastal Zone Management), WFD (Water Framework Directive) and MSFD (Marine Strategy Framework Directive) especially on habitat, eutrophication and hydrographic condition descriptors.

### 4.1 Operational oceanography in coastal waters

### 4.1.1 State-of-the-art

#### Monitoring

Monitoring in the coastal waters has been particularly active in the past decade through both in situ and remote sensing. Comprehensive coastal observatories have been established and maintained in the UK, Germany and some other countries. Integrated monitoring using HF radar, ferrybox, mooring buoy, gliders and satellites have provided huge amounts of observations. The EC has also strongly supported the coastal monitoring infrastructure, e.g. through projects JERICO, JERICO-NEXT and other funding instruments (e.g. European structural funds). Monitoring for commercial purposes also represents a significant data source. However, the value of existing observations in the

**OSD**

doi:10.5194/os-2015-103

**Developing European operational oceanography for Blue Growth**

J. She et al.

Discussion Paper | Discussion Paper | Discussion Paper | Discussion Paper

coastal waters has far from been fully exploited, especially for operational oceanography. First, project-oriented observations have poorly been integrated into operational data flow for forecasting, second, new knowledge generated from the high resolution observations in the coastal waters is still limited.

In the next few years, a large amount of high resolution satellite observations will be available including the ocean colour (Sentinel 3), sediment (FCI from Meteosat Third Generation) and coastal altimetry (Sentinels). In the long-run it is expected that SWOT will provide altimetry sea level in swath and hydrological monitoring of big rivers. This will provide a sustainable monitoring base for operational oceanography in coastal wa-
ters.

Vertical stratification in coastal areas, especially in the river mouths, estuaries and enclosed basins, largely influences the vertical transport of substances as well as their transformation in the pycnoclines, redoxcline and at the water–sediment interface. Thus, high resolution observations through the entire water column to resolve relevant
features and processes in stratified regions have to be applied. The challenge here is to achieve the proper resolution both in time and in space.

**Modelling and forecasting**

There have been two major issues in focus in the past decade: one is how to bridge and couple the global and basin scale forecasting systems with coastal modelling ap-
plications, the other is to integrate the fragmented coastal modelling systems at European scale (She and Buch, 2003). The FP6 project ECOOP was developed with the objective to consolidate, integrate and further develop existing European coastal and regional seas operational observing and forecasting systems into an integrated pan-European system targeted at detecting environmental and climate changes, predicting
their evolution, producing timely and quality assured forecasts, and providing marine information services (including data, information products, knowledge and scientific advices). Such objectives and tasks are now largely taken over by CMEMS. The research

Discussion Paper | Discussion Paper | Discussion Paper | Discussion Paper |

**OSD**

doi:10.5194/os-2015-103

**Developing European operational oceanography for Blue Growth**

J. She et al.

in this area has been identified as a CMEMS research priority – seamless interactions between basin and coastal systems (CMEMS STAC, 2015).

However, many key dynamic processes in the CSW have not been well resolved by the existing forecasting systems developed in ECOOP and CMEMS. This includes coupling between sediment, optics, physical and ecosystem, vertical exchange between atmosphere, water and bottom, bathymetry change, interaction between river and sea waters, small scale features such as sub-mesoscale eddies, river plumes etc., Sediment transport and coastal morphology models have not been included as part of the forecasting system.

Alternatively, the coupled hydrodynamic-wave-sediment models have been developed and used in commercial applications for many years. Some of them are even made available for the public use. It is expected that the existing knowledge and modelling tools for CSW will be integrated into operational systems through close cooperation between the operational oceanography community and the private sector.

### 4.1.2   Research priorities in coastal waters

The long-term goal is to develop an operational oceanography framework which can resolve major marine data and information service issues especially in the CSW. This requires upgrading existing operational coastal ocean forecasting system with new components (e.g., sediment transport, inundation model, marine optics model) and new dynamic processes which are currently missing.

Establishment of operational oceanography addressing CSW is a significant initiative and big step to lift the role of operational oceanography in Blue Growth. This needs support at European scale. EuroGOOS has revised the agreement for membership which now allows a private company to be a formal member. This will largely facilitate the cooperation between the operational community and private sectors. Support from the EC with large-scale projects is essential to ensure the necessary funding for both integration activities and research on new knowledge generation and transformation into operational systems.

**OSD**

doi:10.5194/os-2015-103

**Developing European operational oceanography for Blue Growth**

J. She et al.

Discussion Paper | Discussion Paper | Discussion Paper | Discussion Paper |

The short- to mid-term objective is to build up operational monitoring and forecasting systems in the CSW. Engaging existing monitoring into an operational framework, harvesting new knowledge and developing CSW modelling and forecasting technology are the three major pillars to reach the objective.

Monitoring and data management research: in addition to research recommended in Sect. 2, specific R&D activities are needed: enhance monitoring coordination in cross-board and regional scales, expanding existing HF-radar observing system to cover European coastal seas, engaging research and commercial monitoring activities to be part of the operational dataflow, ensuring delivery of new in situ and satellite observations for operational usage.

New knowledge generation for improving CSW models: new knowledge on key dynamic processes, such as hydrodynamic-sediment-optics-biological interactions, three dimensional current-/sea level-wave interaction, vertical flux exchange between atmosphere, water and sea floor, sub-mesoscale phenomena and interaction between sea and river waters etc., can be obtained by using high resolution in situ and remote sensing data together with modelling tools. The new knowledge harvesting shall aim at improving coastal ocean models.

Modelling and forecasting technology: developing coastal ocean models for the CSW to resolve key dynamic processes in CSW through transferring new knowledge obtained into models, including hydrodynamic-sediment-optics-biological coupling, ocean-wave-ice coupling, improved description of vertical exchange and sub-mesoscale parametrisation, developing sub-kilometric resolution estuary models, coupling between storm surge, wave and inundation models, building up operational monitoring and forecasting capacity for sediment transport, including operational data provision, model development and data assimilation, data assimilation of high resolution observation data: ocean colour, sediment, currents, sea level etc., preparation of high quality input datasets for the CSW forecasting system: high resolution bathymetry, sea floor sedimentation types and updates of such datasets, high resolution weather reanalysis and forecasts at kilometre resolution with riverine inputs.

**OSD**

doi:10.5194/os-2015-103

**Developing European operational oceanography for Blue Growth**

J. She et al.

### 4.1.3 Coastal hazard prediction

Coastal hazards, including hydro-meteorological hazards, coastal erosion, pollution and ecological hazards, are one of the major threats to sustainable development in Blue Growth. Risk management in response to the coastal hazards require improved deterministic and probabilistic predictions in the short-term as well as estimation of historical events and statistics and future projections.

For coastal erosion and pollution: research shall aim at gaining understanding of: (i) processes governing variability in the surface layer (mixed layer turbulence, interactions with air–sea fluxes) and linking surface wave, currents and sediment resuspension and pollutant transportation, (ii) processes in the bottom boundary layer including resuspension that are important, e.g., for the exchange of properties across shelf breaks and for the behaviour of dense sill overflows and better water column optics, (iii) the role of riverine inputs, advection and sedimentation in coastal sediment balance and modelling and predicting coastal sediment balance, (iv) the impact of coastal erosion due to waves and sea level rise. The knowledge obtained from the above should be used to improve predictive sediment and pollutant models. The in situ monitoring of sediment should be enhanced with innovative technology. Operational sediment transportation models should be developed, calibrated and satellite sediment data should be assimilated. The long-term goal of coastal sediment transport research should aim at an operational framework that can support seamless data and information flows for a well-balanced and objective decision-making in ICZM.

For coastal hydro-meteorological hazards: understanding, modelling and prediction of hydro-meteorological hazards such as flooding, storm surge and high seas, developing ensemble and super-ensembles technology for forecasting hydro-meteorological extreme events, developing nowcasting technology by assimilating real time radar, in situ and satellite data into operational models for search and rescue, for civil protection and risk management, coupled weather-ocean-wave-inundation models in the coastal zone should be developed and calibrated.

**OSD**

doi:10.5194/os-2015-103

**Developing European operational oceanography for Blue Growth**

J. She et al.

Title Page

Abstract | Introduction

Conclusions | References

Tables | Figures

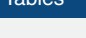 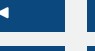

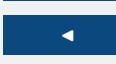 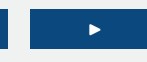

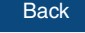 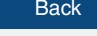 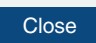

**OSD**

doi:10.5194/os-2015-103

**Developing European operational oceanography for Blue Growth**

J. She et al.

For ecological hazards: understanding, modelling and prediction of ecosystem hazards, integrated forecasting system should be developed for predicting HAB, hypoxia and loss of habitat. New knowledge and understanding on the driving forces and internal mechanisms and evolution of ecological hazards are required. Based on the new knowledge obtained, the operational models can be further optimised so that they are capable of properly simulating the ecological hazard events. Assimilation technology should be used to develop the forecast and pre-warning capacity of the ecological hazard. The research in this area needs to be integrated with R&D activities in Sect. 5 – Operational Ecology.

In the long-term, an operational approach for the integrated coastal service focusing on the coastal zone should be developed. Such an approach will, on the one hand, extend existing coastal and shelf sea forecasting system to coastal zone with higher resolution, on the other hand, develop new, standardized and integrated service tools and products, which feature

– a common framework to bridge CMEMS and national coastal services,

– a seamless coastal forecasting service: model resolution ranging from hundreds of meters to kilometres, high resolution measurements from HF radar, ferrybox, buoys and gliders assimilated. The model system shall resolve challenging processes and features in coastal waters such as currents/sea level-wave-ice interaction, inter-basin and inter-sub-basin exchange, strong density gradients in estuaries, transport of momentum, heat and sediment in very shallow waters etc. Combining modelling and monitoring tools: assimilating, advantages of High Performance Computing are drawn for high resolution climate simulations,

– objective methods of generating indicators for ICZM based on seamless flow of data and information products,

– value-added operational indicator products for public stakeholder use,

- transformation from new knowledge into new operational services such as sediment forecast and coastal morphology forecast,

- transformation from new knowledge into operational information products in pan-European coastal waters, such as rapid mapping of coastal water mass properties (water quality and physical features), dynamic features such as position of river fronts, distribution of eddy energy, position of (semi)permanent coastal currents etc.,

- reconstruction, prediction and projection of the changing coastal environment due to climate change and natural variability.

Potential change of human activities in coastal oceans due to climate change adaptation and mitigation, societal and economic change should be considered and transferred into scenarios for European coastal oceans, such as

- change in offshore exploitation (wind energy, oil and gas etc, some are due to climate change adaptation),

- change in shipping activities (some are due to climate change such as ice melting),

- change in riverine discharge (due to legislation),

- change in land use in the coastal zone,

- change in fishery (due to climate change and fishery management).

The impacts of these scenarios can be projected and assessed by using the tools and products developed for the integrated coastal service.

Another long-term goal is to deepen our understanding on the sub-mesoscale features in coastal and shelf seas. Due to the launch of the SWOT satellite mission after 2020, swath-based altimetry data and hydrological observations will be available. This

**OSD**

doi:10.5194/os-2015-103

**Developing European operational oceanography for Blue Growth**

J. She et al.

Discussion Paper | Discussion Paper | Discussion Paper | Discussion Paper

may lead to enhanced knowledge on the sub-mesoscale features in the coastal waters. Advection and mixing associated with mesoscale and sub-mesoscale oceanic features such as river fronts, meanders, eddies and filaments are of fundamental importance for the exchanges of heat, fresh water and biogeochemical tracers between the surface and the ocean interior, but also exchanges between the open oceans and shelf seas.

The challenges associated with mesoscale and sub-mesoscale variability (between 1–20 km) in the coastal oceans imply therefore high-resolution observations (both in situ and satellite) and multi-sensor approaches. Accordingly, as suggested in CMEMS Service evolution strategy (CMEMS STAC, 2015), multi-platform synoptic experiments have to be designed in areas characterized by intense density gradients and strong mesoscale activity to monitor and establish the vertical exchanges associated with mesoscale and sub-mesoscale structures and their contribution to upper-ocean interior exchanges.

## 4.2 Climate change impacts on the coastal environment

Climate change poses one of the main challenges faced by society in the coming decades, especially to fragile coastal environment. Its impact in many cases is amplified by anthropogenic activities in coastal regions. Operational oceanography community in Europe also provides marine climate service to the society and Blue Growth through further extending its operational monitoring and modelling capacities to climate scale. Considering recent trend in seamless earth system modelling and prediction, weather, ocean and climate research will become more and more integrated. In this section we address research on the coastal ocean climate change adaptation and mitigation related to operational oceanography.

Major research objectives of coastal operational oceanography on climate scales are (i) to provide long-term historical data, including both observations through integrating and re-processing and model reanalysis by assimilating observations into operational models, (ii) to develop operational ocean-ice models for climate modelling and projections, (iii) to identify major climate change signals in the past and future coastal

**OSD**

doi:10.5194/os-2015-103

**Developing European operational oceanography for Blue Growth**

J. She et al.

environment ranging from seasonal to centennial scales and (iv) to assess the impact of climate change and adaptation and mitigation measure on operational scenarios. In addition to improving the climate service quality at national level, these activities will also contribute to the consolidation of Ocean State reports delivered by CMEMS, and to the development of the Copernicus Climate Change Service (C3S). The following research activities have been identified.

### 4.2.1 Short- to mid-term objectives

– Reduction of systematic errors in the reprocessing, modelling and assimilation components for the production of long-term historical data, improved methods to account for representability and sampling observation errors.

– Advancement of operational coastal ocean-ice models for climate modelling: benchmark equilibrium test of operational models to ensure their long-term stability using "free runs", to reduce uncertainties in climate downscaling by optimising downscaling model dynamics and setup. Development of reliable techniques to forecast regional/local sea-level rise including the land-rising term in the ocean climate models, enhance the ocean climate model performance on modelling storm surge events, resolving "skin effect" for more accurate SST modelling.

– Improved understanding of coastal sea-level forcing mechanisms and coupling with the regional variability in climate models, research on relative sea-level trends in relation to future storm tracks and changing storm surges, developing and undertaking a detailed assessment of the extent of coastal erosion in the EU at appropriate temporal and spatial scales, identification of climate variability on stratification and its relation to climate change of other ocean properties.

**OSD**

doi:10.5194/os-2015-103

**Developing European operational oceanography for Blue Growth**

J. She et al.

Discussion Paper | Discussion Paper | Discussion Paper | Discussion Paper

#### 4.2.2 Long-term objectives

- Developing a probabilistic framework and metrics for ocean reanalysis using ensemble and super-ensemble techniques, including inter-comparison, verification, defining and generating probabilistic tailored products for users etc., developing methods to ensure quality, homogeneity and robust uncertainty measures In the long-term time-series reconstructed from data or model reanalysis.

- New methods and diagnostics to evaluate the climate change predictability of the ocean circulation, biogeochemistry and marine ecosystems at basin scale and coastal scale to provide a theoretical basis for the long-term prediction.

- Methodologies to project information about the present ocean state and variability into the future, based on a combination of reanalysis and Earth system models.

## 5 Operational ecology

Timely and regular assessment of the status of the marine environment and its ecosystems is essential for ecosystem-based management in the implementation of EU regulations such as MSFD, Water Framework Directive (WFD), Common Fishery Policy (CFP) and regional conventions etc. Operational Ecology (OE) is the systematic and operational provision of quality assured data and information on the status of marine ecosystems (environment, low trophic and high trophic levels) to stakeholders through integrating research, operations and services (the relationship of OE Research to the Operational-Service is shown in the flowchart in Fig. 2). OE data products are generated from combinations of remote sensing and in situ measurements and marine ecosystem models with data assimilation for the past (reprocessed long-term observation time series and reanalysis), current and recent (analysis and updated rolling reanalysis) and future (short-term/seasonal/decadal forecast and scenario projections). OE information products are value-added and derived from the OE data products,

Discussion Paper | Discussion Paper | Discussion Paper | Discussion Paper | Discussion Paper |

**OSD**

doi:10.5194/os-2015-103

**Developing European operational oceanography for Blue Growth**

J. She et al.

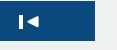
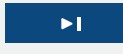
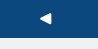
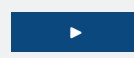
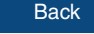
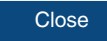

for example GES (Good Environmental Status) criteria and indicators (as defined in the MSFD Common Implementation Strategy) and seasonal/annual marine ecosystem status reports, derived from the OE data products. OE products will make ecosystem-based management more reliable, operational, efficient and timely by providing:

- more frequent updates of environment and ecosystem state,

- more reliable and efficient assessments based on an integrated model-EO approach,

- new capabilities for ecosystem forecast at seasonal to decadal scales,

- flexible operational tools for scenario-based ecosystem management through end-to-end modelling (lower trophic level, e.g. plankton, to higher trophic level, e.g. fish, mammals etc., Rose et al., 2010),

- provision of an operational higher trophic level service for fishery management,

- more reliable forecasts of biohazards such as harmful algal bloom, hypoxia etc.,

- more reliable projections of long-term trend, fluctuation, and regime shift of marine ecosystems.

## 5.1 Enhanced monitoring and forecast capacities for marine ecosystems

Recent scientific developments and breakthroughs have provided a preliminary knowledge base and associated data delivery, models and analysis tools to begin to address the above issues. For example the EU FP7 project OPEC (Operational Ecology) has developed and evaluated ecosystem monitoring tools to help assess and manage the risks posed by human activities on the marine environment, thus improving the ability to predict the "health" of European marine ecosystems. OPEC developed prototype ecological marine forecast systems for European seas (North-East Atlantic, Baltic, Mediterranean and Black Seas), which include hydrodynamics, lower and higher trophic levels

**OSD**

doi:10.5194/os-2015-103

**Developing European operational oceanography for Blue Growth**

J. She et al.

**OSD**

doi:10.5194/os-2015-103

(plankton to fish) and biological data assimilation and made demonstration reanalysis simulations, assessed the effectiveness of the current operational ecosystem monitoring systems and demonstrated the potential to make robust seasonal ecosystem forecasts. In addition the OPEC project has developed an open source web GIS data portal (http://portal.marineopec.eu/) and a model benchmarking tool which allows users to visualize, plot, download and validate large spatial–temporal data sets. Figure 3 shows an example of dynamic viewing of reanalysis and rapid environmental assessment for a user-selected region (marked as square).

Simultaneously, the FP7 OSS2015 project has developed R&D activities with the objective to derive representations of biogeochemical variables from the integration of gliders and floats with EO satellite data into cutting-edge numerical biogeochemical and bio-optical models. There is an expectation that the integrated Atlantic Ocean Observing System (developed through AtlantOS) will increase the number and quality of in situ observations on chemistry, biology and ecology over the next decade. A co-evolution of the data use in assessment and predictive models holds great potential for new products and users.

It is expected that results from these projects as well as similar advances in the field will be transferred to operational services such as CMEMS. The relevant short- to long-term research objectives in this area have been identified in the MSFD Session in the CMEMS Service Evolution and User Uptaking Workshop (Brussels, 2015). They are further evolved in following sections.

### 5.1.1 Short- to mid-term objectives

– Data: increasing the amount of biogeochemical data which can be used for validation and assimilation, through enhanced data sharing, shortening the delivery time and making new observations via innovative instruments e.g. Bio-Argo, extension of existing monitoring capabilities from primary production to plankton.

**OSD**

doi:10.5194/os-2015-103

**Developing European operational oceanography for Blue Growth**

J. She et al.

– Quality assurance: developing a standardized validation method/system for ecosystem model products/variables (particularly related to non-assimilated observations/variables), to identify major weaknesses of existing operational ecological models regarding to needs of ecosystem-based management in national and regional levels (e.g. MSFD).

– Model optimisation: improving existing operational ecological models regarding the weaknesses identified by transferring state-of-the-art biological knowledge into model terms, developing new modules linking optical properties in the near-surface ocean to biomass, improved representation of key processes such as primary production, nutrient uptake, grazing etc. in models resolving the diurnal variability, demonstration of consistent interfacing (nesting, downscaling) between open ocean biogeochemical models and regional/coastal ecosystem models and downstream applications.

– Forecast technology: development of probabilistic (ensemble-based) ecosystem modelling approaches including uncertainty estimation capabilities.

– Multi-data assimilation capabilities (combining state and parameter estimation): combining ocean colour and sub-surface data from relevant ecological observations especially in regional seas, simultaneous assimilation of physical and biological properties.

– Tailored provision of operational products in addition to standard (water temperature, salinity, ice, waves, mixing features, residence time, Chl $a$, oxygen, pH, nutrients, light, plankton biomass) in support of predictive habitat forecasts, for ecological status and fisheries modelling and risk assessment (e.g. invasive species, HABs).

– Rapid environmental and ecosystem assessment: developing an efficient data framework, assimilation and assessment tools to provide a rapid mapping of seasonal or annual marine environment and ecosystem states. The marine environ-

Discussion Paper | Discussion Paper | Discussion Paper | Discussion Paper

ment and ecosystem states are assessed by a set of GES (Good Environmental Status) indicators derived from the rolling analysis. Since the GES indicators are used in MSFD assessment, the assessment means an "operational approach" for the sustainable management of ecosystem resources which provides solid basis
⁵ for the future MSFD assessment.

### 5.1.2   Long-term objectives

- Monitoring: more homogenous biogeochemical monitoring network in Europe, Improved methodologies for supplying operational information on sources of nutrients and pollution/chemicals to the oceans (e.g., CDOM, underwater noise and
¹⁰ plastic/paraffin etc.).

- Modelling: improved description of benthic-pelagic coupling on short-term (seasonal) and long-term (decadal) scale, identification of good initial conditions, improving representation of biological cycles in sea ice, including optical properties of sea ice and vertical migration of nutrients in sea ice.

¹⁵ - New capabilities for ecosystem projections at seasonal (to decadal) scales.

### 5.2   Climate variability and marine ecosystems

While seasonal variability is the most prominent mode in the natural variability of marine ecosystems these seasonal cycles are also modulated by longer term climate signals. Ecosystem-based management also normally has time scales from seasonal
²⁰ to decadal. Therefor it is essential to understand marine ecosystem change on climate scales in order to make good prognostic models. Climate change and direct anthropogenic activities are two major classes of pressures changing the state of the marine ecosystem. The research in this area has been carried out in EU project MEECE. Recent progress has been reviewed by Barange et al. (2014). For the European Seas,
²⁵ the climate change impacts on marine ecosystems were reviewed by EU FP6 project

Discussion Paper | Discussion Paper | Discussion Paper | Discussion Paper | Discussion Paper |

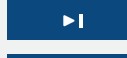
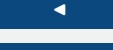
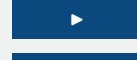

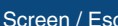

Interactive Discussion

CLAMER (ESF Marine Board, 2011). Future research priorities have also been identified. However, for the OE, what we are interested is the research that can directly improve the predictive capability on marine ecosystems.

– Impacts of long-term change of water temperature, salinity, mixing features, upwelling, coastal circulation, riverine inputs, ice conditions, inter-basin exchange and their impacts on basin and coastal ecosystems, investigate potential relations between climate change pressures and ecosystem long-term change such as regime shift.

– Investigate if increasing atmospheric supply of nutrients could potentially offset the reduced oceanic vertical supply.

– Couple regional climate change scenarios with river basin, nutrient transfer and coastal ecosystem models, to test the interacting effects of global climate change with scenarios of regional socio-economic change, better understanding of the possible responses of coastal ecosystems to changing riverine nutrient loads, flooding and warming.

– Improving the understanding and prediction of ocean acidification by combining in situ and satellite observations.

– Impacts of sea level rise and land vertical movement on change of shorelines, loss of habitat and coastal ecosystems.

– Understanding predictability of the biogeochemistry and marine ecosystems at basin scale or regional scale: identifying major forcing-dependent predictability signals in marine ecosystems, improved understanding of impacts of long-term change of inter-basin, inter-sub basin and riverine inputs on basin and coastal ecosystems.

**OSD**

doi:10.5194/os-2015-103

**Developing European operational oceanography for Blue Growth**

J. She et al.

**OSD**

doi:10.5194/os-2015-103

**Developing European operational oceanography for Blue Growth**

J. She et al.

### 5.3 Operational Ecology European Experiment – OEEE

Operational ecology is a new and emerging research area. For the moment, provision of a quality assured ecological service on seasonal forecasting, annual assessment, decadal reanalysis and scenario projections of marine ecosystems on an operational basis are non-existent. Major knowledge gaps exist in:

- Processes in understanding and modelling biogeochemical cycle in the regional seas, interaction between low trophic level and high trophic level and benthic ecosystems.

- Data assimilation techniques for biogeochemical parameters that focus on improving long-term forecasts and statistics.

- Forecasting technology on seasonal and longer time scales.

- More accurate modelling and estimation of river nutrient loading, spreading and fate in the sea.

- High trophic level modelling and forecasting technology.

- End-to-end modelling for operational scenario projections.

The gaps in the knowledge base, monitoring networks and product quality are interdependent. Among them, the availability of the observations is the basis for advancing the process understanding, filling the knowledge gaps and quantifying and improving the product quality. On the one hand, operational monitoring systems provide information on the state of the system which allows us to assess model performance in predicting the state of the system and hence improve skill through data assimilation and parameter tweaking etc. On the other hand, filling knowledge gaps requires dedicated process studies, which can be used to develop terms missing and parameterise the processes in the models. This is an arguably pre-cursor R&D that underpins the more applied R&D required for OE. In OPEC, it was found that significant monitoring gaps

exist in the Mediterranean and the Black Sea biogeochemical monitoring networks while relative smaller gaps (in terms of effective spatial coverage) are encountered in the Baltic and North Sea. However, the data availability is still not fit for the purpose of providing operational seasonal forecast and rapid environment assessment on an annual basis.

Without timely and sufficient observations, the OE product quality cannot be verified at basin scales, not to mention further optimisation of the modelling systems which needs observations for calibration and process studies. On the other hand, rational sampling schemes (sampling frequency and locations) are essential for making better forecasts. It was found that optimal re-location of the existing North Sea buoys can increase the explained North Sea temperature variability by a factor of two (She et al., ODON final report). In OPEC, it was found that changing sampling frequency from weekly to daily of a ferrybox line in the Aegean Sea can increase the explained chl $a$ variability from 35 to 96.5 %.

In order to build up a quality assured European capacity to deliver the OE service, an "Operational Ecology European Experiment – OEEE" is required. This would serve as part of the mid- to long-term research element of European OE. The goal of the OEEE is to integrate as many as possible existing observations and advanced modelling technologies to develop and demonstrate OE showcases in European regional seas. This can be reached through six research activities:

- to establish a comprehensive database by integrating existing European marine monitoring components (as described in Sect. 2) for testbed studies,

- to develop new knowledge and related new/improved parameterisations on key biogeochemical processes in the models, new field experiments should be designed to collect necessary observations for the dedicated OE research,

- to make breakthrough in advancing ocean-ice-ecosystem full-scale models by transferring the new knowledge obtained to model processes,

**OSD**

doi:10.5194/os-2015-103

**Developing European operational oceanography for Blue Growth**

J. She et al.

Discussion Paper | Discussion Paper | Discussion Paper | Discussion Paper |

**OSD**

doi:10.5194/os-2015-103

– to understand the ecosystem model behaviour in a probabilistic framework, aiming at generating unbiased ensembles (regarding to ecosystem reality) for the dedicated model system,

– to improve the quality of the forcing data from atmosphere deposits, riverine inputs and physical ocean, as well as better description and parameterisation of the forcing terms,

– to generate OE products to assimilate as much as possible observations into the improved model system in an ensemble framework. The products will cover different temporal scales. For historical reanalysis and rapid ecosystem mapping, the physical-biogeochemical-ecosystem model (both lower and high trophic level) will be used, for the future outlook and scenario projections, end-to-end models will be used as necessary through coupling with the physical-biogeochemical-ecosystem models. The products will serve ecosystem-based management at European scale to serve dedicated stakeholders such as EEA, ICES, Regional Conventions and Member State environmental agencies for the implementation of WFD, MSFD and CFP etc.

The research priorities in OEEE are as follows:

– development of modelling and forecasting techniques for GES assessment and operational fishery management:

  a. developing fully coupled atmosphere–ocean-ice-biogeochemical-IBM-food web models for ecosystem outlook at seasonal to decadal scales in a probabilistic framework, optimising the computational performance and resolution to resolve all important pressures and processes,

  b. improving understanding of nutrient cycles and key parameterizations in forecast models,

  c. improving understanding of ecosystem predictability from synoptic to seasonal scales,

**Developing European operational oceanography for Blue Growth**

J. She et al.

Discussion Paper | Discussion Paper | Discussion Paper | Discussion Paper | Discussion Paper

OSD

doi:10.5194/os-2015-103

Developing European
operational
oceanography for
Blue Growth

J. She et al.

Discussion Paper | Discussion Paper | Discussion Paper | Discussion Paper

    d. improving data assimilation for operational forecasts of marine ecological hazards and seasonal to decadal outlooks,

    e. further development of flexible end-to-end models/tools for scenario-based services, improving coupling between end-to-end models and operational models,

– improving the understanding of impacts of atmospheric deposit, riverine inputs, discharges from vessels and bottom resuspension on the ecosystem states, better description of the forcing terms and improved forcing data quality for ecological models,

– integrating existing marine observation components (as mentioned in Sect. 2) for operational ecology through data assimilation, model calibration and validation,

– designing and recommending new monitoring activities in order to reduce major uncertainties in operational ecology products,

– providing a preoperational demonstration (multi-model ensemble approach) of Rapid Environmental Assessment with comprehensive data assimilation, seasonal to decadal forecasting/projections for ecosystem components with high predictability and fisheries service.

## 6   Summary and discussion

In this paper, major research challenges on European operational oceanography are identified for the four knowledge areas as (i) European ocean observations: improving the cost-effectiveness of the marine observation systems, integration of European marine observations, developing innovative monitoring technology and optimal design of sampling schemes and harmonised use of the observational infrastructure, (ii) modelling and Forecasting Technology: development of Unified Ocean system Models,

data assimilation and forecasting technology for seamless modelling and prediction, (iii) Coastal Operational Oceanography: development of operational coastal oceanography to resolve sub-mesoscale features and shallow coastal waters, integrating science, observations and models for new knowledge generation and operational system
development and (iv) operational ecology: development of operational ecology to resolve entire marine ecosystems from physical ocean to high trophic level food-web at relevant scales. The first two areas "European ocean observations" and "Modelling and Forecasting Technology" are the basic instruments for European operational oceanography. The last two "Coastal Operational Oceanography" and "Operational Ecology"
aim at developing corresponding operational approaches.

    For European ocean observation research, further advancement of the existing operational observation infrastructure remains to be the primary focus, especially on biogeochemical variable, extreme events and sub-mesoscale features. On-going use and new development of observation capacities, such as Sentinels, FCI, Cryosat2, SWOT,
ITP, bio-Argo etc., will make it possible to generate new and/or better scientific understanding, operational applications, products and services.

    For Modelling and Forecasting Technology, recent efforts on advancing the ocean system models for next generation super-computing architectures, coupled modelling, innovative data assimilation approaches and probabilistic forecasting will provide es-
sential elements for building up the UOMs. The operational ocean system models will be calibrated to meet the energy and mass conservation conditions so that they can be used for climate predictions and projections.

    Operational oceanography is a major developer and provider of marine services for supporting Blue Growth, and also an important instrument to integrate and sustain Eu-
ropean marine science. Through integrating and standardising the fragmented knowledge, monitoring and modelling activities into an operational framework with a common value chain, the operational approach will provide a sustained development and service platform and significantly improve efficiency, quality and delivery time of the current services. Two new areas are identified for developing operational approaches

Discussion Paper | Discussion Paper | Discussion Paper | Discussion Paper

**OSD**

doi:10.5194/os-2015-103

**Developing European operational oceanography for Blue Growth**

J. She et al.

for the implementation of integrated coastal zone management and ecosystem-based management, based on the scientific state-of-the-art and user needs. Three specific pillars are used when developing the operational framework for a relative new area: integration of existing capacities into an operational framework, identification and filling key knowledge gaps and transferring the new knowledge into operational instruments. Furthermore, it is essential for the operational oceanography community to work together with the private sector, stakeholders and the non-operational research community when developing the operational frameworks in the targeted areas.

For Coastal Operational Oceanography, new knowledge is needed to understand the interactions between the atmosphere, ocean, wave, ice, sediment, optics and ecosystem, and between river, land and coast waters, as well as between sub-mesoscale and other scales. By integrating the new knowledge, new observations and coastal marine system models into an operational framework, an operational modelling and forecasting capacity will be established for the shallow coastal waters.

For the development of an operational approach for marine ecology, new knowledge is needed in understanding ecosystem functions such as nutrient cycle, benthic-pelagic interaction, lower-high trophic coupling, the response of the marine ecosystem to external pressures caused by climate change and human activities, and the transport of chemicals and pollutants exported from the atmosphere, rivers and vessels.

European research on operational oceanography will be sustained by national activities for improving the national marine products and services, regional networking activities such as ROOSes, regional-EU joint research activities such as BONUS-163, European program Horizon 2020 especially themes "Bio-economy, marine and maritime" and "Climate, environment and sustainable development", and CMEMS.

The research and development in EuroGOOS is coherent with the vision of IOC: "Strong scientific understanding and systematic observations of the changing world ocean climate and ecosystems shall underpin sustainable development and global governance for a healthy ocean, and global, regional and national management of risks and opportunities from the ocean (IOC, 2014)". It significantly contributes to the

**OSD**

doi:10.5194/os-2015-103

**Developing European operational oceanography for Blue Growth**

J. She et al.

four IOC high level objectives in the IOC Medium-Term Strategy 2014–2021 document, i.e. (i) healthy ocean ecosystems and sustained ecosystem services, (ii) effective early warning systems and preparedness for ocean-related hazards, (iii) increased resilience to climate change and variability and enhanced safety, efficiency and effectiveness of
all ocean-based activities through scientifically-founded services, adaptation and mitigation strategies and (iv) enhanced knowledge of emerging ocean science issues.

Furthermore European operational oceanography research will actively contribute to the relevant international organisations and programs such as WMO, JCOMM, GEO, GOOS, GCOS, GODAE-Oceanview and YOPP etc. Their scientific strategies and im-
plementation plans provide multiple focus issues and also references for European operational oceanography research and services. Due to the limit of space, the detailed relation between European operational oceanography research and the international programmes is not analysed in this paper.

It should be mentioned that the knowledge areas and research priorities identified
are not exhaustive. Some important scientific areas, such as monitoring and forecasting at global scale and ice infested waters and satellite operational oceanography, are not addressed sufficiently in this paper. These issues can be found in scientific strategy documents in programmes contributing to the operational oceanography development such as CMEMS, YOPP and PEEX (Pan Euro-Asia Experiment) etc. and review papers
e.g. by Le Traon et al. (2015).

It is anticipated that more and more service areas for the Blue Growth, climate change adaptation and ecosystem-based management will adopt an "Operational Approach" which shares a similar operational service value chain. In many cases integration of marine and sectorial information products is needed for such an approach, which
requires that operational oceanography community to work together with the sectorial marine service providers, facilitators, stakeholders and end users. The European operational oceanography community will be dedicated to identify, develop and cultivate the Operational Approaches for marine services in the corresponding socio-economic areas through address new research challenges in the emerging service areas.

**OSD**

doi:10.5194/os-2015-103

**Developing European operational oceanography for Blue Growth**

J. She et al.

Discussion Paper | Discussion Paper | Discussion Paper | Discussion Paper |

*Acknowledgements.* The authors thank EuroGOOS Working Groups, Task Teams, ROOSes, CMEMS STAC and several European and national R&D projects, for their inputs and comments to the paper.

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

**Table A1.** List of acronyms.

| | |
|---|---|
| 3-D/4-DVAR | Three/Four Dimensional VARiational method |
| ADCP | Acoustic Doppler Current Profiler |
| AtlantOS | Optimizing and Enhancing the Integrated Atlantic Ocean Observing System |
| BFM | Biogeochemical Flux Model |
| BG | Blue Growth |
| C3S | Copernicus Climate Change Service |
| CFP | Common fishery Policy |
| CDOM | Colored Dissolved Organic Matter |
| CLAMER | Climate change and European Marine Ecosystem Research project |
| CMEMS | Copernicus Marine Service |
| COSYNA | Coastal Observing System for Northern and Arctic Seas project |
| CPU | Central Processing Unit |
| CSW | Costal Shallow Waters |
| EDIOS | European Directory of the Ocean-Observing System project |
| ECMWF | European Centre Medium-range Weather Forecast |
| ECO3M | Mechanistic Modular Ecological Model |
| ECOOP | European COastal-shelf sea OPerational observing and forecasting system project |
| ECOSMO | ECOSystem MOdel |
| EEA | European Environment Agency |
| EGO | Everyone's Glider Observatories |
| EMODnet | European Marine Observation and Data Network |
| EnKF | Ensemble Kalman Filter |
| EnVAR | Ensemble-Variational method |
| EO | Earth Observation |
| EOOS | Sustained European Ocean Observing System |
| ERGOM | Ecological ReGional Ocean Model |
| ERSEM | European Regional Seas Ecosystem Model |
| ESF | European Science Foundation |
| EU | European Union |
| EuroGOOS | European Global Ocean Observing System |
| FCI | Flexible Combined Imager |
| FixO3 | Fixed point Open Ocean Observatory network project |
| FP | EC Framework Program |

**OSD**

doi:10.5194/os-2015-103

Developing European operational oceanography for Blue Growth

J. She et al.

| | |
|---|---|
| GCOS | Global Climate Observing System |
| GDP | Gross Domestic Production |
| GEO | Group of Earth Observations |
| GES | Good Environmental Status |
| GIS | Geographic Information System |
| GMES | Global Monitoring for Environment and Security |
| HAB | Harmful Algae Bloom |
| HBM | HIROMB-BOOS Model |
| HELCOM | The Baltic Marine Environment Protection Commission |
| HF | High Frequency |
| HPC | High Performance Computing |
| HYCOM | HYbrid Coordinate Ocean Model |
| ICES | International Council for the Exploration of the Sea |
| ICZM | Integrated Coastal Zone Management |
| ITP | Ice-Tethered Profiler |
| JCOMM | Joint Technical Commission for Oceanography and Marine Meteorology |
| JERICO | Joint European Research Infrastructure network for COastal observatories project |
| MFC | Monitoring and Forecasting Centres |
| MITGCM | MIT General Circulation Model |
| MONALISA2 | Securing the Chain by Intelligence at Sea project |
| MSFD | Marine Strategy Framework Directive |
| NEMO | Nucleus for European Modelling of the Ocean |
| NORWECOM | NORWegian ECOlogical Model system |
| ODON | Optimal Design of Observational Networks project |
| OE | Operational Ecology |
| OEEE | Operational Ecology European Experiment |
| OOPS | Object Oriented Programming System project |
| OPEC | OPerational Ecology project |
| OSE | Observing System Experiment |
| OSS2015 | Ocean Strategic Services beyond 2015 project |
| OSSE | Observing System Simulation Experiment |
| OSPARCOM | Convention for the Protection of the Marine Environment of the North-East Atlantic |

Discussion Paper | Discussion Paper | Discussion Paper | Discussion Paper |

# OSD

doi:10.5194/os-2015-103

**Developing European operational oceanography for Blue Growth**

J. She et al.

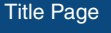

Title Page

Abstract | Introduction

Conclusions | References

Tables | Figures

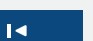 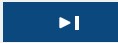

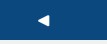 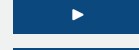

**OSD**

doi:10.5194/os-2015-103

| | |
|---|---|
| PDAF | Parallel Data Assimilation Framework |
| PSyKAl | Parallel System, Kernel and Algorithm |
| RASM | Regional Arctic System Model |
| ROMS | Regional Ocean Modeling System |
| ROOS | Regional Operational Oceanography System |
| R&D | Research and Development |
| SANGOMA | Stochastic Assimilation for the Next Generation Ocean Model project |
| SeaDataNet | Pan-European infrastructure for ocean and marine data management project |
| SEEK | Singular Evolutive Extended Kalman filter |
| SIMD | Single Instruction Multiple Data |
| SMOS | Soil Moisture Ocean Salinity |
| SST | Sea Surface Temperature |
| SSS | Sea Surface Salinity |
| SWOT | Surface Water and Ocean Topography |
| TB | Tera Byte |
| $T/S$ | Temperature/Salinity |
| UAM | Unified Atmospheric Model |
| UEM | Unified Earth system Model |
| UM | Unified Model |
| UOM | Unified Ocean system Model |
| YOPP | Year of Polar Prediction |
| WFD | Water Framework Directive |
| WMO | World Meteorological Organisation |

**Developing European operational oceanography for Blue Growth**

J. She et al.

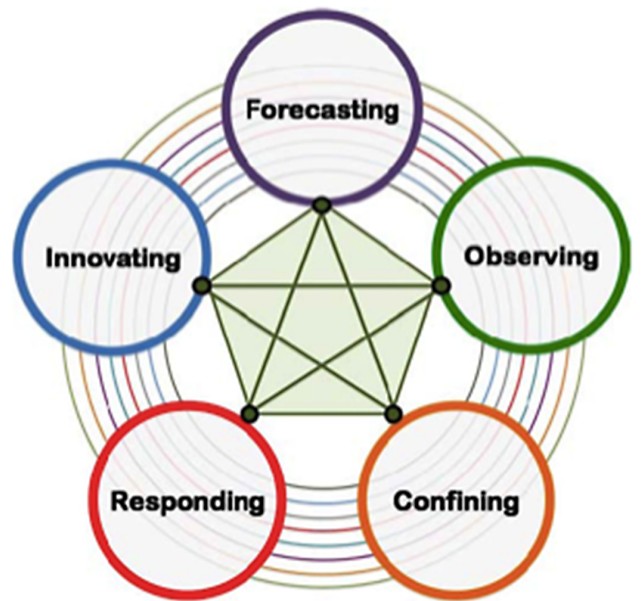

**Figure 1.** Schematic of grand challenges from the "Earth System Science for Global Sustainability". Source: ICSU (2010).

Discussion Paper | Discussion Paper | Discussion Paper | Discussion Paper

**OSD**

doi:10.5194/os-2015-103

**Developing European operational oceanography for Blue Growth**

J. She et al.

**OE Research**
- Design technical requirement of next generation
- Optimisation of models and monitoring networks
- Forecasting technology
- Ecosystem assimilation
- Integrated assessment method

Requirement for next generation

**OE Service**
- Product dissemination
- User requirement, feedback & verification
- User uptake
- Service and service data requirement
- Service evolution strategy

**OE Operation**

**Observations**
- Quality control
- Pre-processing
- Data management

**Reconstruction**
- Objective analysis
- Analysis/ initialisation
- Re-processing
- Long-term Reanalysis
- Updated annual reanalysis

**Forecast and projections**
- Short-term forecast
- Seasonal forecast
- Decadal forecast
- Scenarios

**Product management**
- User interface
- System monitoring
- Value added post-processing

**Figure 2.** Flowchart of the operational ecology.

Discussion Paper | Discussion Paper | Discussion Paper | Discussion Paper | Discussion Paper

**OSD**

doi:10.5194/os-2015-103

**Developing European operational oceanography for Blue Growth**

J. She et al.

**OSD**

doi:10.5194/os-2015-103

**Developing European operational oceanography for Blue Growth**

J. She et al.

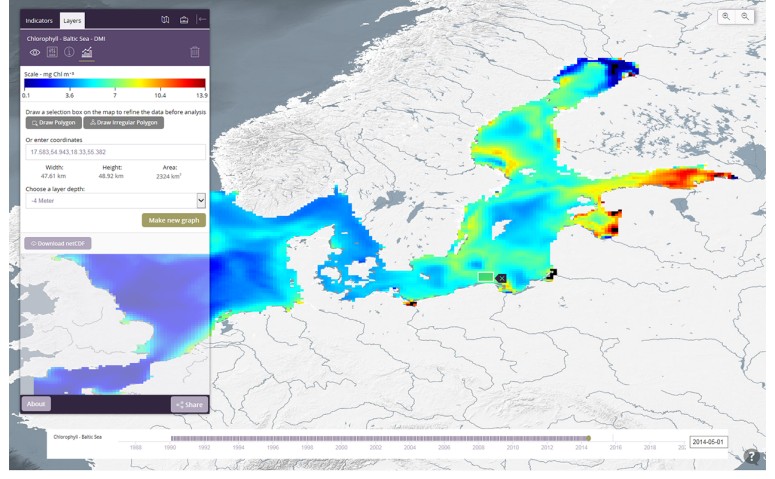

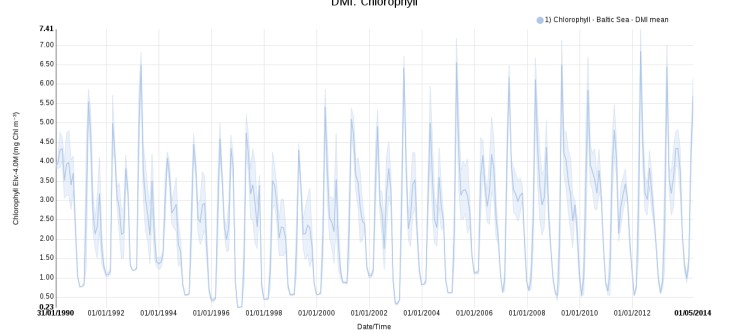

**Figure 3.** OPEC Rapid Environment Assessment and multi-decadal biogeochemical reanalysis: an example of Baltic Sea chl *a*. Upper panel: OPEC data portal for extracting OE products, lower panel: monthly mean chl *a* time series during January 1990–May 2014 at a selected rectangular polygon shown in the upper panel. The shadow area shows the monthly standard deviation.