# Peer review of "Developing European operational oceanography for Blue Growth, climate change"

_Ocean Science, 2015_

## Referee Comment (RC1) · Anonymous Referee #1 · 1 Mar 2016

Written by leading scientists from EuroGOOS and the European operational oceanography community (http://www.eurogoos.eu/), this whitepaper presents in a clear, concise and well structured way knowledge gaps and scientific challenges that shall be addressed by the community for the next decade.

This paper is clearly written to arouse the awareness of the national and European research funding agencies to these challenges that request their support. However, the interest of this paper goes further this initial objective as it provides a rather fair and extensive description of the current operational oceanography landscape and helps identifying and explaining several emerging, often misunderstood but already important concepts such as operational ecology.

[Figure]

That being said, by definition, such a whitepaper defends subjective positions even if I would say most of the presented positions are largely shared by the community. As a consequence reviewing such paper is also a difficult task since reviewers' position is for sure as subjective as the authors' one.

I will therefore limit my review to two points that according to me are not fair enough or don't reflect the reality.

The first comment is about section 3.1 "Model development". This is a clear example of a subjective position. In response to the concept of "Unified Earth Modelling system" introduced by WMO, the Authors introduce the concept of "Unified Ocean Modelling system" or UOM. They define it as an ocean model able to simulate in a seamless and coupled way all the different subsystem models : waves, hydrodynamics –and not only ocean- , sediment transport, water quality and ecosystem. The authors present the concept of UOM as a brand new idea and clearly developed this long section (∼150 lines between slide 16 to slide 20) in order to defend the vision of the CMEMS SAC members about the future developments of the models used in the framework of the Copernicus Marine Environment Monitoring Service (CMEMS). However, models such as for instance the Belgian COHERENS model (http://odnature.naturalsciences.be/coherens/about) or the Portugese MOHID model (http://www.mohid.com/) already fulfil the UOM definition. Being continuously developed and improved for the last 25 years, both models are used in operational oceanography applications and have worldwide user communities dealing with local, coastal, regional and basin-scale applications for short term forecast as well as for climate applications. As a consequence, the paper will really gain in pertinence and credibility if the Authors could refine their vision instead of just inserting the following poor disclaimer: "It is worthwhile to mention that the model development in the CMEMS strategy mainly focuses on the evolution of the existing global and basin scale operational models, new emerging models (. . .) and models for downstream services (. . .) have not been sufficiently addressed in the strategy" (slides 20 and 21).

The second major comment is about the introductory paragraph of "Modelling and forecasting" in section 4.1.1 "Operational oceanography in the coastal waters – State-of-the-Art " in slides 29 and 30. The Authors suggest there that CMEMS plays a leading role to consolidate and homogenise the fragmented coastal forecasting community and has therefore taken over objectives and tasks of the ECOOP project. According to me, this is false. The ECOOP project has been closed out 5 years ago. Since them, some activities developed in the ECOOP project have been taken over by EuroGOOS ROOSes; other have been continued at Institute's level; the remaining ones have simply disappeared by lack of financial support. For these 5 years, CMEMS and the previous MyOcean projects have plaid no significant role to structure the coastal forecasting community. Therefore, I would really advice the Authors to better explain their meaning. In particular, I'm wondering whether the sentence "Such objectives and tasks are now largely taken over by CMEMS" (line 27, slide 29) has just been added to better sell the subjective position of the CMEMS SAC members : "The research in this area has been identified as a CMEMS research priority - seamless interactions between basin and coastal systems" (lines 1 and 2, slide 30) or whether this sentence finds its origin in the delegation agreement Mercator-Océan has received from the European Commission to implement the Copernicus Marine Environment Monitoring Service.

To conclude my review, this article is a position paper listing the challenges that request the support of the main national and European funding agencies for the next decade. In case Ocean Science publishes such articles, I would advice the editor to accept it for publication provided the final manuscript takes into account the two comments I've pointed out.
* * *

---

## Editor Comment (EC1) · H. Bonekamp (Editor) · 24 Mar 2016

This paper is a well structured and well formulated, providing a clear though general overview of Operational Oceanography in Europe mainly from the perspectives of European Global Ocean Observing System (EUROGOOS). The topics although not truly scientific in nature perfectly map on the objective of this special issue to explain the current state of the art and developments of Operational Oceanography in Europe. System components such as the observing network and the modeling and forecasting capability are explained in general terms and related projects. Key references are made to project acronyms and recent publications, providing the reader an interesting general perspective and overview. Acronyms are spelled out. Furthermore, the paper

focus on two major application areas: coastal operational oceanography and operational (marine) ecology. In these areas, the 'blue' and 'green' services are expected to substantially contribute to (economic) growth.

Small remarks:

'JPI-Ocean' is neither referenced nor in the acronym list.

Is 'BONUS-163' sufficiently explained? Generally, the list of acronyms may need to be checked again on completeness.

―――――――――――――――――――

---

## Author Comment (AC2) · 8 Apr 2016

Thanks for the comment. I went through the text to check if the acronym has been used properly. The following changes in the text will be Applied:

P9, L6: ERIC -> ERIC (European Research Infrastructure Consortium) P9, L9: JPI-Oceans -> JPI-Oceans (The Joint Programming Initiative Healthy and Productive Seas and Oceans) P13, L9: ISO -> ISO (International Organization for Standardization) P13, L17: DG-MARE –> DG-MARE (The Directorate-General for Maritime Affairs and Fisheries) P22, L20: NCAR -> NCAR (National Center for Atmospheric Research) P45, L20: IBM -> IBM (Individual Based Model) P48, L22: BONUS-163 -> BONUS (The joint Baltic Sea research and development programme) P48, L25: IOC -> IOC (Intergovernmental Oceanographic Commission) P55, adding a line "GODAE Global Ocean Data Assimilation Experiment"

---

## Referee Comment (RC2) · Anonymous Referee #2 · 5 May 2016

This position paper has been drafted by leading scientists, all deeply involved in the development of various components of operational oceanography at national, regional, and global scales.

The paper is quite extensive and despite the fact that the various sections are well written, certain sections confuse the reader, probably due to jumps from one topic to another: as for example on page 9, lines 10-27.

The paper has to be shortened and be more focused on those aspects that indeed concern practical topics of operational oceanography, and not on the theoretical topics. This is because several activities mentioned in the paper, as for example those regarding the ocean monitoring and data management, are not directly related to operational

oceanography (SeaDataNet, EMODNET, ICES).

As clearly stated by the terms of the service operation, operational oceanography at coastal waters is not addressed by CMEMS. Following ECOOP, there was almost no serious-coordinated attempt to harmonize coastal operational oceanography on a systematic way, in order to align the models for example with the new developments implemented at regional levels of CMEMS. Any effort on coastal operational oceanography, particularly on forecasting, was mainly based on national interest.

The extended references for an ocean UOM system are indeed a vision. However, the operational oceanographic community at present, need to be consolidated with common tools that will ensure the harmonization with CMEMS in the coming years, to reproduce the correct sea conditions, assimilating in a common way in-situ and remote sensing observations.

Within the section on research priorities in the coastal waters, on page 30, lines 23-28, the text on EuroGOOS new membership does not add any important issue to the scope of the paper, therefore these lines must be deleted.

The paper as is in its present form is quite extensive, therefore I propose: a) To reduce it, excluding those paragraphs that do not add important or significant information about the needs and the aspects of operational oceanography. b) Present in a different and separate ways the vision for both global and regional operational oceanography, which despite their similarities are quite different. c) Provide further information about the coastal scale operational oceanography and expand on what has been achieved so far, as very little information is given in the paper at the present.

---

## Author Comment (AC3) · 28 May 2016

The authors thank the useful comments give by the reviewer. We divided the comments into 6 questions and reply to them one by one. Q1: This position paper has been drafted by leading scientists, all deeply involved in the development of various components of operational oceanography at national, regional, and global scales. The paper is quite extensive and despite the fact that the various sections are well written, certain sections confuse the reader, probably due to jumps from one topic to another: as for example on page 9, lines 10-27. Reply: The paragraph mentioned here (p9, L10-27) is about satellite oceanography. As pointed out by the reviewer, the topic is shifted from in-situ observations to satellite. I proposal to use following sentences from

L10, to avoid such a feeling of jump of topics: Besides the progresses made in in-situ marine observations, the development of satellite oceanography in the last two decades has also been significantly advanced in the last two decades and become a major component of operational oceanography , as documented by Le Traon et al. (2015). Satellites provide real time and regular, global, high spatial and temporal resolution observation of key ocean variables that are essential to constrain ocean models through data and/or to serve downstream applications.

Q2: The paper has to be shortened and be more focused on those aspects that indeed concern practical topics of operational oceanography, and not on the theoretical topics. This is because several activities mentioned in the paper, as for example those regarding the ocean monitoring and data management, are not directly related to operational oceanography (SeaDataNet, EMODNET, ICES). Reply: In view of the authors, the future development of operational oceanography will not only be limited in the Operational Oceanography (OO) community, the development of OO will have to engage some key non-operational players in the monitoring and modelling community. One issue is the coordination and integration of the OO observations with non-operational observing components: on the one hand, this will provide much larger datasets, both near real time and offline data, for operational modelling, assimilation and forecasting. On the other hand, the OO community integrates data with models which maximize the value of observations from environmental, fishery and research monitoring activities. Considering this is a strategic paper, the authors think that we should keep the relevant issues described in the paper, which will be difficult to reach if the paper is shortened. Q3: As clearly stated by the terms of the service operation, operational oceanography at coastal waters is not addressed by CMEMS. Following ECOOP, there was almost no serious-coordinated attempt to harmonize coastal operational oceanography on a systematic way, in order to align the models for example with the new developments implemented at regional levels of CMEMS. Any effort on coastal operational oceanography, particularly on forecasting, was mainly based on national interest. Reply: In Europe, it is expected that the future coastal OO will be covered by following components: 1. National operational agencies w hich are providing some coastal operational services, e.g., storm surge, hydrological forecast, oil spill drift forecast, agitation, inundation/flooding forecast etc. This part of the coastal OO will be expanded due to the increasing user needs, improved monitoring and forecasting capacities; 2. Private companies which have extensive experiences in the coastal services. It is expected that some of their service areas will be transformed into an operational approach, either through cooperation with operational agencies or run the service by themselves; 3. EC funded coastal service, e.g., through CMEMS. Through enhanced resolution, two-way nesting or unstructured grid, regional models can also provide certain type of coastal services based on modelling and forecast of hydro-biogeochemical parameters. Similar experiments have been carried out in ECOOP. However, due to the complexity in the coastal waters, significant research on the estuary-coast-sea interaction will be needed to fill in the knowledge gaps. 4. European ROOSs are the main coordination body which covers the coastal OO. It can be expected that the ROOSs will play a more active role in the integration of coastal OO in the future. So, the integration of coastal OO in Europe is a complicated multi-actor program, which landscape has not yet well defined. This is also why the paper does not go into details for the future solution. However, as suggested in Q6c by the reviewer, that the coastal OO section will be rewritten to reflect more state-of-the-art and potential approaches for building up the coastal services. Q4: The extended references for an ocean UOM system are indeed a vision. However, the operational oceanographic community at present, need to be consolidated with common tools that will ensure the harmonization with CMEMS in the coming years, to reproduce the correct sea conditions, assimilating in a common way in-situ and remote sensing observations. Reply: The authors agree. The short-term research objective of the ocean modelling is to optimize the deterministic models, data assimilation scheme etc, which is consistent with what is mentioned by the reviewer. The paper has promoted community models (e.g. NEMO is mentioned). Q5: Within the section on research priorities in the coastal waters, on page 30, lines 23-28, the text on EuroGOOS new membership does not add any important issue to

the scope of the paper, therefore these lines must be deleted. Reply: We do believe EuroGOOS new membership for private companies will strengthen the cooperation between operational agencies and private companies, especially in developing future coastal operational oceanography. However, as this is just an expectation from the authors and the situation in the future can be quite complicated, we would like to be more careful about this statement. Therefore we decide to delete it, as suggested by the reviewer.

Q6: The paper as is in its present form is quite extensive, therefore I propose: a) To reduce it, excluding those paragraphs that do not add important or significant information about the needs and the aspects of operational oceanography. b) Present in a different and separate ways the vision for both global and regional operational oceanography, which despite their similarities are quite different. c) Provide further information about the coastal scale operational oceanography and expand on what has been achieved so far, as very little information is given in the paper at the present. Reply: Q6a: we haven't covered all important issues in OO but we tried to address major important issues in the four knowledge areas. The authors would like to keep the current structure of the paper. Q6b: the suggested extension to cover the global part will make the text further lengthy. Actually, many challenges in global operational oceanography, e.g., monitoring and modelling systems, have not been addressed by the paper. This is the weakness of the paper. Similar to satellite oceanography (not fully addressed in the paper), global OO should be addressed in a separate paper. Nevertheless the authors have decided to focus on regional operational oceanography. Q6c: Coastal OO will be re-written, to add state-of-the-art in the area and to give a more clear picture about what has been done and what are still needed. The modified text of section 4.1.1 is as follows: 4.1.1 State-of-the-art Monitoring: Monitoring in the coastal waters has been particularly active in the past decade through both in situ and remote sensing. Comprehensive coastal observatories have been established and maintained in the UK, Germany and some other countries. Integrated monitoring using HF radar, ferrybox, mooring buoy, shallow water Argo floats, gliders, integrated sensors and satellites have
provided huge amounts of observations. An important feature is that many of these in-situ datasets have high spatial or temporal resolution, which reveals mesoscale and sub-mesoscale features in coastal waters and processes of estuary-coast-sea interaction. The EC has also strongly supported the development and integration of coastal monitoring infrastructure, e.g. through projects JERICO, COMMONSENSE, JERICO-NEXT and other funding instruments (e.g. European structural funds). Monitoring for commercial purposes also represents a significant data source. However, the value of existing observations in the coastal waters has far from been fully exploited, especially for operational oceanography. First, project-oriented observations have poorly been integrated into operational data flow for forecasting;, second, new knowledge generated from the high resolution observations in the coastal waters is still limited; third, the coastal observations have rarely been assimilated into operational models in near real time mode.

[revised manuscript text omitted]

Please also note the supplement to this comment:
http://www.ocean-sci-discuss.net/os-2015-103/os-2015-103-AC3-supplement.pdf

---

## Author Response (AR2)

**Reply to Comments from Reviewers and Editor**

Please notify that all the comments and reply are regarding to the pdf version in the discussions, which page and line numbers are not the same in the marked-up version.

Reply to Editor's comment:

**Editor's comments:**

This paper is a well structured and well formulated, providing a clear though general overview of Operational Oceanography in Europe mainly from the perspectives of European

Global Ocean Observing System (EUROGOOS). The topics although not truly scientific in nature perfectly map on the objective of this special issue to explain the current state of the art and developments of Operational Oceanography in Europe. System components such as the observing network and the modeling and forecasting capability are explained in general terms and related projects. Key references are made to project acronyms and recent publications, providing the reader an interesting general perspective and overview. Acronyms are spelled out. Furthermore, the paper focus on two major application areas: coastal operational oceanography and operational (marine) ecology. In these areas, the 'blue' and 'green' services are expected to substantially contribute to (economic) growth. Small remarks: 'JPI-Ocean' is neither referenced nor in the acronym list. Is 'BONUS-163' sufficiently explained? Generally, the list of acronyms may need to be checked again on completeness.

**Reply:**

Thanks for the comment. I went through the text to check if the acronym has been used properly. The following changes in the text will be applied:

P9, L6: ERIC -> ERIC (European Research Infrastructure Consortium)

P9, L9: JPI-Oceans -> JPI-Oceans (The Joint Programming Initiative Healthy and Productive

Seas andOceans)

P13,L9: ISO->ISO(InternationalOrganizationforStandardization)

P13, L17: DG-MARE –> DG-MARE (The Directorate-General for Maritime Affairs and

Fisheries)

P22, L20: NCAR -> NCAR (National Center for Atmospheric Research)

P45, L20: IBM -> IBM (Individual Based Model)

P48, L22: BONUS-163 -> BONUS (The joint Baltic Sea research and development programme)

P48, L25: IOC -> IOC (Intergovernmental Oceanographic Commission)

P55, adding a line "GODAE Global Ocean Data Assimilation Experiment"

**Review from Reviewer #1**

Anonymous Referee #1

Written by leading scientists from EuroGOOS and the European operational oceanography community (http://www.eurogoos.eu/), this whitepaper presents in a clear, concise and well structured way knowledge gaps and scientific challenges that shall be addressed by the community for the next decade. This paper is clearly written to arouse the awareness of the national and European research funding agencies to these challenges that request their support. However, the interestof this papergoes further thisinitial objective as itprovides a ratherfair and extensive description of the current operational oceanography landscape and helps identifying and explaining several emerging, often misunderstood but already important concepts such as operational ecology.

That being said, by definition, such a whitepaper defends subjective positions even if I would say most of the presented positions are largely shared by the community. As a consequence reviewing such paper is also a difficult task since reviewers' position is for sure as subjective as the authors' one. I will therefore limit my review to two points that according to me are not fair enough or don't reflect the reality. The first comment is about section 3.1 "Model development". This is a clear example of a subjective position. In response to the concept of "Unified Earth Modelling system" introduced by WMO, the Authors introduce the concept of "Unified Ocean Modelling system" or UOM. They define it as an ocean model able to simulate in a seamless and coupled way all the different subsystem models : waves, hydrodynamics –and not only ocean- , sediment transport, water quality and ecosystem. The authors present the concept of UOM as a brand new idea and clearly developed this long section (∼150 lines between slide 16 to slide 20) in order to defend the vision of the CMEMS SAC members about the future developments of the models used in the framework of the Copernicus Marine Environment Monitoring Service (CMEMS). However, models such as for instance the Belgian COHERENS model (http://odnature.naturalsciences.be/coherens/about) or the Portugese MOHID model (http://www.mohid.com/) already fulfil the UOM definition. Being continuously developed and improved for the last 25years, both models are used in operational oceanography applications and have worldwide user communities dealing with local, coastal, regional and basin-scale applications for short term forecast as well as for climate applications. As a consequence, the paper will really gain in pertinence and credibility if the Authors could refine their vision instead of just inserting the following poor disclaimer: "It is worthwhile to mention that the model development in the CMEMS strategy mainly focuses on the evolution of the existing global and basin scale operational models, new emerging models (...) and models for downstream services (...) have not been sufficiently addressed in the strategy" (slides 20 and 21).

The second major comment is about the introductory paragraph of "Modelling and forecasting" in section 4.1.1 "Operational oceanography in the coastal waters – State-of-the-Art " in slides 29 and 30. The Authors suggest there that CMEMS plays a leading role to consolidate and homogenise the fragmented coastal forecasting community and has therefore taken over objectives and tasks of the ECOOP project. According to me, this is false. The ECOOP project has been closed out 5 years ago. Since them, some activities developed in the ECOOP project have been taken over by EuroGOOS ROOSes; other have been continued at Institute's level; the remaining ones have simply disappeared by lack of financial support. For these 5 years, CMEMS and the previous MyOcean projects have plaid no significant role to structure the coastal forecasting community. Therefore, I would really advice the Authors to better explain their meaning. In particular, I'm wondering whether the sentence "Such objectives and tasks are now largely taken over by CMEMS" (line 27, slide 29) has just been added to better sell the subjective position of the CMEMS SAC members : "The research in this area has been identified as a CMEMS research priority - seamless interactions between basin and coastal systems" (lines 1 and 2, slide 30) or whether this sentence finds its origin in the delegation agreement Mercator-Océan has received from the European Commission to implement the Copernicus Marine Environment Monitoring Service. To conclude my review, this article is a position paper listing the challenges that request the support of the main national and European funding agencies for the next decade. In case Ocean Science publishes such articles, I would advice the editor to accept it for publication provided the final manuscript takes into account the two comments I've pointed out.

**Reply to Review #1**

Thank you for your good comments. Sorry for a little delayed reply as I just had two BONUS proposals submitted yesterday.

In general, I agree with your view on the two issues, but with some reservations on point 1.

**First comment:**

The vision in the paper is to develop European operational UOMs in i) pan-European scale, ii) Arctic-N. Atlantic Scale, for all ocean variables and all scales. Currently all model systems, either CMEMS model system or estuary-coast-sea model systems (e.g., COHERNS, MOHID, Delfta3D, MIKE system etc) have their own strengths and weaknesses, and only fulfil part of the seamless modelling requirements (here seamless means spatiotemporal-parameter). But it is true that it may cause confusion if the paper refers too much to CMEMS. Modelling is a general issue. So I have modified original text to:

Pages 20/21

Emerging modelling areas: the future UOM needs integration and extensions of current European modelling capacity in spatial-temporal scale and parameter dimensions. It is worthwhile to mention that the model development in the CMEMS strategy mainly focuses on the evolution of tThe existing global and basin scale operational models (ocean–sea ice-wave-biogeochemistry) can be evolved to resolve estuary and straits, while existing estuary-coastal-sea models can be extended to cover multi-basins., Nnew emerging models such as sediment transport and high trophic level models, and models for downstream services such as coastal inundation model, unstructured grid models— need to be further matured and developed and integrated with existing operational systems for operational applications. have not been sufficiently addressed in the strategy. In addition to the model development, comprehensive verification studies should be made especially for the ecological models and models in Arctic models in order to understand their drawbacks. of the models. For the ice model, mesoscale sea ice rheology will be needed to describe lead dynamics of the ice. More discussions on the development of marine ecosystem models can be found in Sect. 5 – Operational Ecology.

**Second comment:**

I agree with your comment: CMEMS is mainly part of continuation of MERSEA and MYOCEAN focusing on Global and Basin scales; ECOOP is for the coastal-shelf sea part. Although CMEMS has done some extension to the coast but has not taken over what ECOOP has developed (e.g. models for 15 high resolution coastal areas). The "seamless interactions between basin and coastal systems" as one of the research priority for CMEMS Service Evolution is not from originally delegation agreement, but to address coastal user needs in downstream services.

So the original text will be modified to:

Pages 29/30

Such objectives and tasks should be further addressed, extended to resolve the estuary-coast-sea interaction and developed into an operational framework through integration into basin-scale operational systems. are now largely taken over by CMEMS. Recently tThe research in this area has been identified as a CMEMS research priority – seamless interactions between basin and coastal systems (CMEMS STAC, 2015).

**Review from Review #2**

Anonymous Referee #2

This position paper has been drafted by leading scientists, all deeply involved in the
development of various components of operational oceanography at national, regional, and
global scales. The paper is quite extensive and despite the fact that the various sections are
well written, certain sections confuse the reader, probably due to jumps from one topic to
another: as for example on page 9, lines 10-27. The paper has to be shortened and be more
focused on those aspects that indeed concern practical topics of operational oceanography,
and not on the theoretical topics. This is because several activities mentioned in the paper, as
for example those regarding the ocean monitoring and data management, are not directly
related to operational oceanography (SeaDataNet, EMODNET, ICES). As clearly stated by
the terms of the service operation, operational oceanography at coastal waters is not addressed
by CMEMS. Following ECOOP, there was almost no serious-coordinated attempt to
harmonize coastal operational oceanography on a systematic way, in order to align the models
for example with the new developments implemented at regional levels of CMEMS. Any
effort on coastal operational oceanography, particularly on forecasting, was mainly based on
national interest. The extended references for an ocean UOM system are indeed a vision.
However, the operational oceanographic community at present, need to be consolidated with
common tools that will ensure the harmonization with CMEMS in the coming years, to
reproduce the correct sea conditions, assimilating in a common way in-situ and remote
sensing observations. Within the section on research priorities in the coastal waters, on page
30, lines 2328, the text on EuroGOOS new membership does not add any important issue to
the scope of the paper, therefore these lines must be deleted. The paper as is in its present
form is quite extensive, therefore I propose: a) To reduce it, excluding those paragraphs that
do not add important or significant information about the needs and the aspects of operational
oceanography. b) Present in a different and separate ways the vision for both global and
regional operational oceanography, which despite their similarities are quite different. c)
Provide further information about the coastal scale operational oceanography and expand on
what has been achieved so far, as very little information is given in the paper at the present.

**Reply to Reviewer #2**

The authors thank the useful comments given by the reviewer. We divided the comments into 6 questions and reply to them one by one.

**Q1**: This position paper has been drafted by leading scientists, all deeply involved in the development of various components of operational oceanography at national, regional, and global scales. The paper is quite extensive and despite the fact that the various sections are well written, certain sections confuse the reader, probably due to jumps from one topic to another: as for example on page 9, lines 10-27.

**Reply**:

The paragraph mentioned here (p9, L10-27) is about satellite oceanography. As pointed out by the reviewer, the topic is shifted from in-situ observations to satellite. I proposal to use following sentences from L10, to avoid such a feeling of jump of topics:

   Besides the progresses made in in-situ marine observations,  satellite oceanography  has also been significantly advanced in the last two decades and become a major component of operational oceanography as documented by Le Traon et al. (2015). Satellites provide real time and regular, global, high spatial and temporal resolution observation of key ocean variables that are essential to constrain ocean models through data and/or to serve downstream applications.

**Q2**: The paper has to be shortened and be more focused on those aspects that indeed concern practical topics of operational oceanography, and not on the theoretical topics. This is because several activities mentioned in the paper, as for example those regarding the ocean monitoring and data management, are not directly related to operational oceanography (SeaDataNet, EMODNET, ICES).

**Reply**:

In view of the authors, the future development of operational oceanography will not only be limited in the Operational Oceanography (OO) community, the development of OO will have to engage some key non-operational players in the monitoring and modelling community. One issue is the coordination and integration of the OO observations with non-operational observing components: on the one hand, this will provide much larger datasets, both near real time and offline data, for operational modelling, assimilation and forecasting. On the other hand, the OO community integrates data with models which maximize the value of observations from environmental, fishery and research monitoring activities.

Considering this is a strategic paper, the authors think that we should keep the relevant issues
described in the paper, which will be difficult to reach if the paper is shortened.

**Q3**: As clearly stated by the terms of the service operation, operational oceanography at
coastal waters is not addressed by CMEMS. Following ECOOP, there was almost no serious-
coordinated attempt to harmonize coastal operational oceanography on a systematic way, in
order to align the models for example with the new developments implemented at regional
levels of CMEMS. Any effort on coastal operational oceanography, particularly on
forecasting, was mainly based on national interest.

**Reply**:

In Europe, it is expected that the future coastal OO will be covered by following components:

1. National operational agencies which are providing some coastal operational services, e.g.,
storm surge, hydrological forecast, oil spill drift forecast, agitation, inundation/flooding
forecast etc. This part of the coastal OO will be expanded due to the increasing user needs,
improved monitoring and forecasting capacities;
2. Private companies which have extensive experiences in the coastal services. It is expected
that some of their service areas will be transformed into an operational approach, either
through cooperation with operational agencies or run the service by themselves;
3. EC funded coastal service, e.g., through CMEMS. Through enhanced resolution, two-way
nesting or unstructured grid, regional models can also provide certain type of coastal services
based on modelling and forecast of hydro-biogeochemical parameters. Similar experiments
have been carried out in ECOOP. However, due to the complexity in the coastal waters,
significant research on the estuary-coast-sea interaction will be needed to fill in the
knowledge gaps.
4. European ROOSs are the main coordination body which covers the coastal OO. It can be
expected that the ROOSs will play a more active role in the integration of coastal OO in the
future.

So, the integration of coastal OO in Europe is a complicated multi-actor program, which
landscape has not yet well defined. This is also why the paper does not go into details for the
future solution. However, as suggested in Q6c by the reviewer, that the coastal OO section
will be rewritten to reflect more state-of-the-art and potential approaches for building up the
coastal services.

**Q4**: The extended references for an ocean UOM system are indeed a vision. However, the
operational oceanographic community at present, need to be consolidated with common tools
that will ensure the harmonization with CMEMS in the coming years, to reproduce the correct
sea conditions, assimilating in a common way in-situ and remote sensing observations.

**Reply**:

The authors agree. The short-term research objective of the ocean modelling is to optimize the deterministic models, data assimilation scheme etc, which is consistent with what is mentioned by the reviewer. The paper has promoted community models (e.g. NEMO is mentioned).

**Q5**: Within the section on research priorities in the coastal waters, on page 30, lines 23-28, the text on EuroGOOS new membership does not add any important issue to the scope of the paper, therefore these lines must be deleted.

**Reply**:

We do believe EuroGOOS new membership for private companies will strengthen the cooperation between operational agencies and private companies, especially in developing future coastal operational oceanography. However, as this is just an expectation from the authors and the situation in the future can be quite complicated, we would like to be more careful about this statement. Therefore we decide to delete it, as suggested by the reviewer.

**Q6**: The paper as is in its present form is quite extensive, therefore I propose: **a)** To reduce it, excluding those paragraphs that do not add important or significant information about the needs and the aspects of operational oceanography. **b)** Present in a different and separate ways the vision for both global and regional operational oceanography, which despite their similarities are quite different. c) Provide further information about the coastal scale operational oceanography and expand on what has been achieved so far, as very little information is given in the paper at the present.

**Reply**:

Q6a: we haven't covered all important issues in OO but we tried to address major important issues in the four knowledge areas. The authors would like to keep the current structure of the paper.

Q6b: the suggested extension to cover the global part will make the text further lengthy. Actually, many challenges in global operational oceanography, e.g., monitoring and modelling systems, have not been addressed by the paper. This is the weakness of the paper. Similar to satellite oceanography (not fully addressed in the paper), global OO should be addressed in a separate paper. Nevertheless the authors have decided to focus on regional operational oceanography.

Q6c: Coastal OO will be re-written, to add state-of-the-art in the area and to give a more clear picture about what has been done and what are still needed. The modified text of section 4.1.1 is as follows:

[revised manuscript text omitted]
.: S. Siiriä O. Vähä Piikki B. Hackett, N. M. Kristensen, H. Engedahl, E. Blockley, A. Sellar, P. Lagemaa, J. Ozer, S. Legrand, P. Ljungemyr and L. Axell: Uncertainty estimation for operational ocean forecast products - A Multi-Model Ensemble for the North Sea and the Baltic Sea. Ocean Dynam.ics, doi:DOI 10.1007/s10236-015-0897-8, 2015

ICSU: Earth System Science for Global Sustainability: The Grand Challenges. International Council for Science, Paris, 2011.

IOC: Medium-Term Strategy 2014-2021, published by UNESCO for IOC, France, September 2014.

Johannessen, J. A., Le Traon, P- Y., Robinson, I., Nittis, K., Bell, M. J., Pinardi, N., and P- Y. Le Traon, I. Robinson, K. Nittis, M. J. Bell, N. Pinardi, and P. Bahurel P.: Marine Environment and Security for the European Area (MERSEA) - Towards operational oceanography, American Meteorological Society, B. Am. Meteorol. Soc., 87, pp. 1081-1090, doi:10.1175/BAMS-87-8-1081, 2006.

Le Traon, P- Y., Bonekamp, H., Antoine, D., Bentamy, A., Breivik, L. A., Chapron, B., P- Y. H. Bonekamp, D. Antoine, A. Bentamy, L.A. Breivik, B. Chapron, G. Corlett G., Dibarboure, G., DiGiacomo, P., Donlon, C., Faugère, Y., Gohin, F., Kachi, M., G. Dibarboure, P. DiGiacomo, C. Donlon, Y. Faugère, F. Gohin, M. Kachi, Font, J., Girard-Ardhuin, F., Johannessen, J. A., Lambin, J., Lagerloef, G., Larnicol, G., Le Borgne, P., Lindstrom, E., Leuliette, E., Maturi, E., Martin, M., 10 Miller, L., Mingsen, L., Morrow, R., Reul, N., Rio, M. H., Roquet, H., Santoleri, R., and Wilkin, J.: J. Font,

F. Girard Ardhuin, J. A. Johannessen, J. Lambin, G. Lagerloef, G. Larnicol, P. Le Borgne, E. Lindstrom, E. Leuliette, E. Maturi, M. Martin, L. 10 Miller, L. Mingsen, R. Morrow, N. Reul, M.H. Rio, H. Roquet, R. Santoleri and J. Wilkin: Use of satellite observations for operational oceanography: recent achievements and future prospects, Community paper - GODAE OceanView Symposium, J. Oper. Oceano. Vol. 8, No. S1, s12–s27, http://dx.doi:.org/10.1080/1755876X.2015.1022050. 2015.

Maslowski, W., J. Clement Kinney, J., M. Higgins, M., and A. Roberts, A.: The future of arctic sea ice, Annu.al Rev.iew of Earth and Pl.anetary Sc.iences, 40, 625-654, 2012.

Nittis, K. and EuroGOOS Board: EuroGOOS Strategy 2014-2020, EuroGOOS publication, Brussels, 2014.

Oke, P. R., and P. Sakov, P.: Assessing the footprint of a regional ocean observing system. Journal of Marine Systems, -105, 30-51, 2012.

Perez, B., Brouwer, R., Beckers, J., Paradis, D., Balseiro, C., Lyons, K., Cure, M., Sotillo, M. G., Hackett, B., Verlaan, M., and Fanjul, E. A.R. Brouwer, J. Beckers, D. Paradis, C. Balseiro, K. Lyons, M. Cure, M. G. Sotillo, B. Hackett, M. Verlaan, and E. A. Fanjul: ENSURF: multi-model sea level forecast – implementation and validation results for the IBIROOS and Western Mediterranean regions. Ocean Sci., 8, 211–226, doi: 10.5194/os-8-211-2012, 2012.

Poulsen, J. W., P. Berg, P., and K. Raman, K.: "Better Concurrency and SIMD On The HIROMB-BOOS-MODEL (HBM) 3D Ocean Code", Chapter 3 in: "High Performance Parallelism Pearls: Multicore and Many-core Programming Approaches, edited by" Jim Jeffers, J. and James Reinders, J., (eds.). Morgan Kaufmann Publishing, USA, 2014.

Prandle, D., J. She, J., and J. Legrand, J.: Operational Oceanography - the Stimulant for Marine Research in Europe., Iin: Marine Science Frontiers for Europe, edited by Wefer, G., Lamy, F., and Mantoura, F. (eds), Marine Science Frontiers for Europe. Springer-Verlag, Berlin-Heidelberg-New York-Tokyo, pp. 161-171, 2003.

Rose, K. A., A., Allen, J. I., Artioli, Y., Barange, M., Blackford, J., Carlotti, F., Cropp, R., Daewel, U., Edwards, K., Flynn, K., Hill, S. L., HilleRisLambers, R., Huse, G., Mackinson, S., Megrey, B., Moll, A., Rivkin, R., Salihoglu, B., Schrum, C., Shannon, L., Shin,Y- J., Smith, S. L., Smith, C., Solidoro, C., John, M. S., and Zhou, M.:J. I. Allen, Y. Artioli, M. Barange, J. Blackford, F. Carlotti, R. Cropp, U. Daewel, K. Edwards, K. Flynn, S. L. Hill, R. HilleRisLambers, G. Huse, S. Mackinson, B. Megrey, A. Moll, R. Rivkin, B. Salihoglu,

C. Schrum, L. Shannon, Y J. Shin, S. L. Smith, C. Smith, C. Solidoro, M. S. John and M. Zhou: End-To-End models for the analysis of marine ecosystems: Challenges, issues, and next steps. Marine and Coastal Fisheries, 2, 115-130, 2010

She, J.: Analysis on research priorities for European operational oceanography, Proceedings of the 7th EuroGOOS Confererence, 28–30 October, 2014, Lisbon, 2015.

She, J. and Buch, E.: Integrated marine science in European shelf sea and adjacent waters, in: Building the European Capacity in Operational Oceanography, edited by Dahlin, H., Flemming, N. C., Nittis, K., and Petersson, S. E., Elsevier publisher, Amsterdam, The Netherlands, 285-290, 2003.

She, J., Høyer, J. L., and Larsen, J.: Assessment of sea surface temperature observational networks in the Baltic Sea and North Sea. J. Marine. Syst., 65, 314-335, 2007.

She J.: Analysis on research priorities for European operational oceanography, Proceedings of the 7th EuroGOOS Confererence, Lisbon. 2015. (submitted)

Shukla, J.: Seamless Prediction of Weather and Climate: A New Paradigm for Modeling and Prediction Research. US NOAA Climate Test Bed Joint Seminar Series NCEP, Camp Springs, Maryland, 2009.

Stanev, E. V., Schulz-Stellenfleth, J., Staneva, J., Grayek, S., Seemann, J. and Petersen, W.: Coastal observing and forecasting system for the German Bight – estimates of hydrophysical states, Ocean Sci., 7, 569-583, doi: 10.5194/os-7-569-2011, 2011.

Stanev, E. V., Ziemer, F., Schulz-Stellenfleth, J., Seemann, J., Staneva, J., and Gurgel, K. –W.: Blending Surface Currents from HF Radar Observations and Numerical Modeling: Tidal Hindcasts and Forecasts. J. Atmos. Oceanic Technol., 32, 256–281, 2015.

Turpin, V., Remy, E., and Le Traon, P. -Y.: How essential are Argo observations to constrain a global ocean data assimilation system?, Ocean Sci. Discuss., 12, 1145-1186, doi:10.5194/osd-12-1145-2015, 2015.

WMO: Seamless prediction of the earth system: from minutes to months, WMO publications-1156, 2015

Zijl, F.,  Verlaan, M., and  Gerritsen, H.: Improved water-level forecasting for the
Northwest European Shelf and North Sea through direct modelling of tide, surge and non-
linear interaction. Ocean Dynam., 63, 823-847, 2013.

[Figure]

Figure 1. Schematic of grand challenges from the 'Earth System Science for Global Sustainability'.

Source: ICSU (2010)

[Figure]

Figure 2. Flowchart of the Operational Ecology

[Figure]

Figure 3 OPEC Rapid Environment Assessment and multi-decadal biogeochemical reanalysis: an example of Baltic Sea chl-a. Upper panel: OPEC data portal for extracting OE

products; lower panel: monthly mean chl-a time series during 1990.01-2014.05 at a selected rectangular polygon shown in the upper panel. The shadow area shows the monthly standard deviation.